
# All-sky Information Content Analysis for Novel Passive Microwave Instruments in the Range from 23.8 GHz up to 874.4 GHz

Verena Grützun[1], Stefan A. Buehler[1], Lukas Kluft[2], Jana Mendrok[3], Manfred Brath[1], and
Patrick Eriksson[3]

[1]Meteorologisches Institut, Fachbereich Geowissenschaften, Centrum für Erdsystem- und Nachhaltigkeitsforschung (CEN),
Universität Hamburg, Bundesstraße 55, 20146 Hamburg, Germany
[2]Max-Planck-Institut für Meteorologie, Hamburg, Germany
[3]Department of Space, Earth and Environment, Chalmers University of Technology, 41296 Gothenburg, Sweden

*Correspondence to:* Verena Grützun (verena.gruetzun@uni-hamburg.de)

**Abstract.**

We perform an all-sky information content analysis for channels in the millimeter/submillimeter wavelength with 24 channels in the region from 23.8 up to 874.4 GHz. Our set of channels corresponds to the instruments ISMAR and MARSS, which are available on the British FAAM research aircraft, and it is complemented by two precipitation channels at low frequencies from Deimos. The channels also cover ICI, which will be part of the MetOp-SG mission. We use simulated atmospheres from the ICON model as basis for the study and quantify the information content with the reduction of degrees of freedom ($\Delta$DOF). The required Jacobians are calculated with the radiative transfer model ARTS. Specifically we focus on the dependence of the information content on the atmospheric composition. In general we find a high information content for the frozen hydrometeors, which mainly comes from the higher channels beyond 183.31 GHz (on average 4.99 for cloud ice and 4.84 for snow). Profile retrievals may be possible for the mass densities and some information about the microphysical properties, especially for cloud ice, can be gained. The information about the liquid hydrometeors comes from the lower channels and is comparably low (2.36 for liquid cloud water and 1.81 for rain). There is little information about the profile or the microphysical properties. The Jacobians for a specific cloud hydrometeor strongly depend on the atmospheric composition. Especially for the liquid hydrometeors they even change sign in some cases. However, the information content is robust. For liquid hydrometeors it slightly decreases in the presence of any frozen hydrometeor, for the frozen hydrometeors it slightly decreases in the presence of the respective other frozen hydrometeor. The overall results with regard to the frozen hydrometeors in principle also hold for the ICI sensor. This points to its great ability to observe ice clouds from space on a global scale with a good spatial coverage in unprecedented detail.

## 1   Introduction

In the last years, passive millimeter/sub-millimeter wavelength measurements of the cloudy sky from space have gained increasing attention. Especially frozen clouds are in the focus of such measurements. And this with a good reason, because clouds are important factors in the climate system. For decades they have contributed to the largest uncertainties to estimating



the Earth's changing energy budget (Boucher et al., 2013). Also, the assimilation of the cloudy sky in numerical weather fore-casting becomes increasingly important (Guerbette et al., 2016). To constrain the estimation of the future development of the climate system and to assimilate the cloudy sky into the weather forecast reliable global observations of clouds are required. Passive millimeter/sub-millimeter wavelength measurements have a great potential to fill that gap.

Many studies have investigated the performance of setups, which employ channels in the range from 5 GHz up to 874 GHz. For example, Di Michele and Bauer (2006) focus on channels between 5 and 200 GHz. They find different suitable frequency bands for rain over ocean, snow over land and ocean, and clouds over ocean and suggest several channels covering these frequency ranges for global and multi-seasonal applications. Jiménez et al. (2007) investigated an instrument with twelve channels around the 183, 325 and 448 GHz water vapour lines and the 234, 664 and 874 GHz window channels. A five-receiver

instrument dropping one of the two highest channels proved to be equally powerful in a mid-latitude scenario as the all-receiver instrument, but for tropical scenarios the highest channel reduced the error for very thin and high clouds. Also, new studies investigate the potential to assimilate microwave sounding data from geostationary satellites into numerical forecast models to further improve the forecast models (Duruisseau et al., 2017).

     There are already very successful missions ongoing, which, amongst other things, observe clouds from space. A well-known

instrument, which is observing the atmosphere from space for decades now, is the Advanced Microwave Sounding Unit B (AMSU-B, Weng et al. (2003); Zhao and Weng (2002)) and its successor, the Microwave Humidity Sounder (MHS, Bonsignori (2007)). AMSU-B operates with five channels in the range from 89 GHz to 183.31 GHz and MHS with five channels in the range from 89 GHz up to 190 GHz, respectively. Although the instruments are primarily designed as humidity sounders, as a side product they also allow for an observation of the ice water path (column integrated ice water mass), rain rate and snow

water equivalent.

     In the near future, the Meteorological Operational Satellite - Second Generation (MetOp-SG, Pica et al. (2012)) with the new Ice Cloud Imager (ICI) will be launched. The principle of ICI is explained in the CloudIce mission proposal for ESA's Earth Explorer 8 (Buehler et al., 2012, 2007). ICI has in total 11 channels in the range from 183.31 GHz to 664.0 GHz and will provide several ice retrievals including the ice water path and the cloud ice effective radius. It will be flown together with

the MicroWave Imager (MWI), which has 18 channels in the range from 18.7 GHz up to 183.31 GHz (see e.g. Accadia et al., 2013, for detailed information about ICI and MWI). The inclusion of the low channels allows for precipitation retrievals.

     In recent years, also the potential of hyper-spectral sensors in the millimeter/submillimeter wavelength region is explored for clear-sky (Mahfouf et al. (2015)) and cloudy-sky (Birman et al. (2017)) conditions. Birman et al. (2017) find that the information content on hydrometeors can be significantly increased by using a hyper-spectral sensor, but also depends on the

assumed microphysical properties of the frozen hydrometeors.

     The different hydrometeor types have different effects on the measurement channels. Several studies focused on the influence of clouds and precipitation on AMSU-like channels around 89, 150 and 183.31 GHz (e.g. Hong et al., 2005; Sreerekha et al., 2008, and references therein). It was found that high level clouds with high cloud tops cause a brightness temperature depression in the channels with frequencies greater than 150 GHz, and that low level clouds only little affect the 183.31 GHz channel

because the largest sensitivity of that channel is too high up in the atmosphere (Burns et al., 1997; Bennartz and Bauer, 2003).





For the same reason, the surface emissivity does not contribute to the signal in these channels. The channel at 89 GHz on the other hand is influenced by altostratus liquid clouds (Muller et al., 1994). Furthermore it is very sensitive to the surface emissivity. Even though the channel at 150 GHz is also a window channel, it shows much less sensitivity to the surface because the region with highest sensitivity to changes in the atmospheric column is located in the lower troposphere above the surface

(Bennartz and Bauer, 2003; Hong et al., 2005). Also the Megha-Tropiques mission (*megha* is the Sanskrit word for clouds, tropiques the French word for tropics, Desbois et al., 2002; Karouche et al., 2012) allows an ice cloud content profile retrieval from the Microwave Analysis and Detection of Rain and Atmospheric Systems (MADRAS sensor, Defer et al. (e.g. 2014)) with channels at 89 GHz and 157 GHz.

Greenwald and Christopher (2002) found that precipitating cold clouds give a much stronger signal in channels near

183.31 GHz compared to cold clouds which are not precipitating. They question the applicability of channels near or below 183.31 GHz to gain quantitative estimates of physical properties of non-precipitating ice clouds from space. In fact, it is very likely that the presence of one hydrometeor type affects the observation of another in the passive observation in the millimeter/sub-millimeter range, because the signal, which is observed at the top of the atmosphere by the satellite is a result of the interaction of the radiation with each atmospheric component present in the pathway. In this article, we specifically

focus on this effect in detail. In the following, we study the information content of passive microwave measurements of clouds from space with specific focus on the cloudy atmosphere. We investigate whether it depends on the combinations of cloud and precipitation hydrometeors within the atmospheric column how much information we get, as the results from Greenwald and Christopher (2002) suggest. To include higher channels, which may be suitable to detect ice microphysical properties, we chose the setup of the instruments MARSS (Microwave Airborne Radiometer Scanning System, McGrath and Hewison

(2001)) and ISMAR (International Sub-millimeter Microwave Airborne Radiometer, Fox et al. (2017)) and complement them by two low channels at 23.8 GHz and 50.1 GHz from Deimos (Dual-frequency Extension to In-flight Microwave Observing System, Hewison (1995)). These instruments cover a large range of microwave channels from 23.8 GHz up to 874.4 GHz (see Sect. 5.1), including the ICI channels, and part of MWI. Thus we can put our focus on the potential of novel instruments operating at frequencies higher than 183 GHz to robustly observe ice, but also include precipitation, which is observed with

the channels lower than 183,GHz.

Since it is impossible to have full knowledge of the real atmosphere, we chose to base our investigations on high-resolution model data from the ICOsahedral Non-hydrostatic model (ICON model, Dipankar et al. (2015); Heinze et al. (2017)), which employs the two-moment microphysics by Seifert and Beheng (2006). We use the reduction of degrees of freedom as a tool to quantify the information content of a measurement with regard to a certain hydrometeor. For this purpose we require Jacobians,

which we explicitly calculate with the Atmospheric Radiative Transfer Simulator (ARTS, Buehler et al. (2005); Eriksson et al. (2011)). We first use an idealised mean profile to do a conceptual study of the mechanisms and then look into a larger set of atmospheric profiles from ICON with the full set of channels and with the channel set corresponding to ICI to investigate if the results hold for more realistic atmospheres.

In the following, we first introduce the underlying modeling framework Sect. 2. Secondly, we present in detail the mi-

crophysical assumptions for the atmospheric and for the radiative transfer model, which we use in our study, in Sect. 3. Our



framework to quantify the information content is presented in Sect. 4. We explain the choice of an idealised atmospheric profile and of 90 realistic profiles, as well as the selected set of channels in Sect. 5. Our results are presented in Sect. 6. Finally we conclude the article in Sect. 7.

## 2 Models

### 2.1 ICON

We base our study on data from the novel ICOsahedral Non-hydrostatic model (ICON model, e.g. Wan et al., 2013; Dipankar et al., 2015). We use a simulation of a frontal case on 26 April 2013 over West Germany with rapidly increasing cloudiness to a completely overcast situation in the afternoon. Several light to medium rain showers happened during that day, and ice clouds as well as snow in the upper atmospheric layers were found. The simulation has a horizontal resolution of 650 m with 50 hybrid terrain-following vertical height levels up to 22 km. It was performed in the framework of the BMBF project High Definition Clouds and Precipitation for advancing Climate Prediction (HD(CP)[2]) and was provided by the Max-Planck Institute for Meteorology, Hamburg. The simulation complements the measurement campaign HOPE (HD(CP)[2] Observation Prototype Experiment, Macke et al. (2017a)) which took place in April and May 2013 around Jülich in the West of Germany and focused on clouds and model evaluation (e.g. Stamnas et al., 2016; Heinze et al., 2017; Macke et al., 2017b)[1].

### 2.2 Atmospheric Radiative Transfer Simulator ARTS

In order to perform an information content analysis a radiative transfer model is required to simulate the satellite measurements and the respective height-resolved Jacobians based on the atmospheric profiles simulated by ICON. We use the Atmospheric Radiative Transfer Simulator (ARTS, Buehler et al., 2005; Eriksson et al., 2011, version 2.3.296). ARTS is an open source detailed line-by-line radiative transfer model for microwave to thermal infrared radiation, which is capable to simulate polarised radiative transfer in all spatial geometries[2]. ARTS offers analytical Jacobians for trace gas concentration, and semi-analytical Jacobians for temperature. In this ARTS version, Jacobians for hydrometeor parameters are calculated by perturbation, which has higher computational costs compared to analytical computation. Details about the calculation of these Jacobians are given in Sect. 4.1 and the specific setup of ARTS is described in Sect. 3.2.

For the radiative transfer calculations we have to assume a surface emissivity $\epsilon$. We perform our analysis with two different emissivities, one equal to 0.6, which corresponds to an ocean surface, and one equal to 0.9, which corresponds to a land surface. We furthermore assume specular reflection. One should keep in mind though, that in reality $\epsilon$ depends strongly on the specific surface and to a smaller extend also on the channel channel. However, the results differ only little for the different emissivities.

---

[1]Details about the project and the campaign can be found on the project homepage http://hdcp2.eu (last assessed July 2017) or on the data base SAMD (Standardised Atmospheric Measurement Data) homepage hosted at the Integrated Climate Data Center (ICDC) under http://icdc.cen.uni-hamburg.de/1/projekte/ samd.html (last assessed July 2017). A special issue of Atmospheric Chemistry and Physics (ACP) about HOPE has been issued (HD(CP)2 Observational Prototype Experiment (AMT/ACP inter-journal SI), 2014, Eds. S. Buehler and H. Russchenberg).

[2]See www.radiativetransfer.org for documentation and download.





Therefore we use the simplified assumption of a constant emissivity for all channels, and the main part of the results we show in this article will be for the emissivity of the ocean, i.e., $\epsilon = 0.6$.

## 3 Microphysical parameterisations

### 3.1 ICON

ICON uses the two-moment microphysical scheme by Seifert and Beheng (2006), which offers more detailed information about the cloud microphysical properties than the commonly used one-moment bulk schemes. It simulates the mass mixing ratio ($M$) and number mixing ratio ($N$) of cloud liquid water, cloud ice, rain, snow, hail and graupel. Since only very little graupel and hail was found in the simulation, we disregard them in the following. For the atmospheric radiative transfer simulator ARTS (Sect. 2.2) we converted the mass mixing ratios (unit kg kg$^{-1}$) to mass densities (kg m$^{-3}$) by multiplying with the density of
the atmosphere.

    In the following, we refer to liquid cloud water mass density (liquid water content) as LWC, to cloud ice mass density as IWC, to rain mass density as RWC and to snow mass density as SWC. We call the different types hydrometeors, and refer to LWC and RWC as liquid hydrometeors and to IWC and SWC as frozen hydrometeors. Note that even though the ICON model's microphysical parameterisation requires a clear distinction between suspended and precipitating hydrometeors in each
grid box, i.e., between LWC and RWC or IWC and SWC respectively, this distinction can not be made in reality. Nevertheless we will discuss the cloud and precipitating hydrometeors separately in the remainder of the article, always keeping in mind that in reality, there is a smooth transition between the cloud hydrometeors and the precipitating hydrometeors and they can not easily be separated.

    For the simulation of the cloud radiative effect the size distribution and shape of the hydrometeors is of high importance. It is
crucial to match the microphysical parameterisations of the radiative transfer model with those of the atmospheric model. The two-moment scheme from Seifert and Beheng (2006) employs a modified $\Gamma$-distribution with two free parameters as particle size distribution functions for each hydrometeor type. It is defined as:

$$f(m) = A m^\nu \exp\left(-\lambda m^\mu\right), \tag{1}$$

where the independent size parameter is the particle mass $m$. The distribution parameters are $A$, $\nu$, $\lambda$ and $\mu$ and have to be
provided by the scheme. In the actual version of Seifert and Beheng (2006)'s scheme, $\nu$ and $\mu$ are fixed for each hydrometeor type (Table 1) and $A$ and $\lambda$ are calculated prognostically (see Seifert and Beheng (2006) for details of the calculation).

    The size distributions from the two moment scheme for an idealised mean profile (purple) and a set of 90 individual ICON profiles (grey) are shown in Fig. 1 (for the definition of the idealised and the 90 profiles please refer to Sect. 5.2). Note that the distributions are height dependent. They are shown at a height of 550 hPa, where both, IWC and SWC exist in considerable
amounts. The curves illustrate the sum of the distributions for IWC and for SWC, i.e., all frozen hydrometeors. For the mean profile, also the individual distributions for IWC and SWC are shown to illustrate to what extent they overlap. The two peaks, which are evident in the idealised and in some of the 90 profiles, result from the two different types of frozen hydrometeors.





**Table 1.** Distribution parameters for the hydrometeor particles after Seifert and Beheng (2006) and pers. comm. Seifert, 2014.

| Hydrometeor type | $\nu$ | $\mu$ |
|---|---|---|
| LWC | 1 | 1 |
| RWC | 0 | 1/3 |
| IWC | 0 | 1/3 |
| SWC | 0 | 1/2 |
| Graupel | 1 | 1/3 |
| Hail | 1 | 1/3 |

The one at smaller diameters belongs to IWC, the one at larger diameters belongs to SWC. It is important to note that in the two-moment scheme the distinction between IWC and SWC is done via the processes a particle has undergone. A particle is counted as snow if it has for example collided and joined with other hydrometeors (e.g. self-collection or collection of smaller hydrometeors). Single ice crystals are counted as cloud ice. Therefore, in the two moment-scheme cloud ice hydrometeors can be quite large in mass equivalent diameter and overlap with snow.

For comparison, the respective size distributions which result from McFarquhar and Heymsfield (1997) are also shown in Fig. 1. Note that the McFarquhar and Heymsfield (1997) parameterisation is based only on the mass densities of the hydrometeors, i.e., it is a one-moment parameterisation. Also, the scheme employs a common distribution of IWC and SWC instead

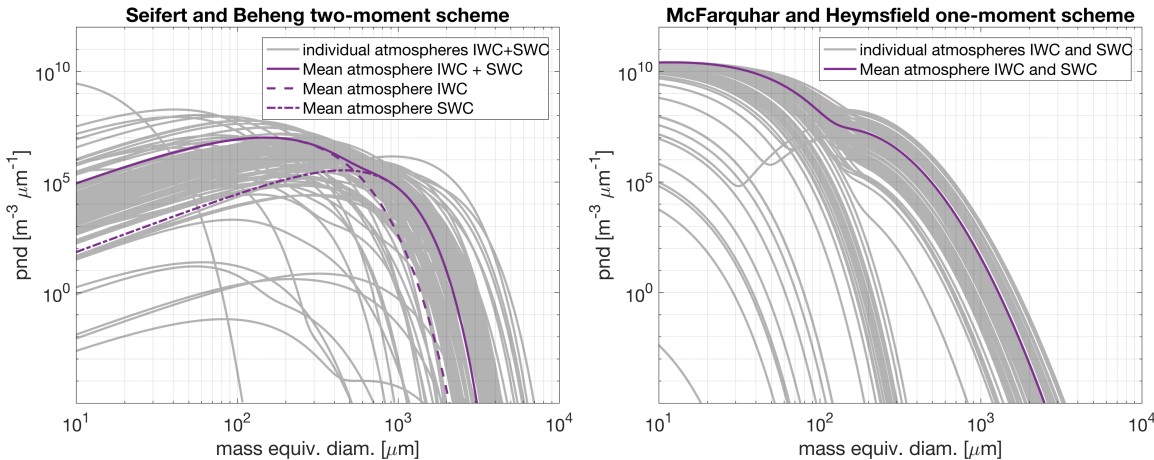

**Figure 1.** Size distributions for the idealised mean profile (purple) and 90 simulated profiles (grey) derived from ICON (left) and from the parameterisation by McFarquhar and Heymsfield (1997) (right) at 550 hPa each. For the two moment scheme the sum of the distributions for IWC and SWC is shown for all profiles, and the individual distributions for IWC (dash-dotted) and SWC (dashed) are shown for the mean profile. For the McFarquhar and Heymsfield scheme, the distribution for the sum of IWC and SWC is shown. The number density is ignored in this parameterisation. No individual distributions for IWC and SWC exist in that case. The same IWCs and SWCs stemming directly from the ICON simulation and at the height 550 hPa have been used in both cases.



of separate ones as used in the two-moment scheme. In the one-moment scheme, the distinction between IWC and SWC is done by setting a threshold for the size. For example, a frozen hydrometeor in McFarquhar and Heymsfield (1997) is counted as snow (large ice particle) if its mass equivalent diameter is larger than $100\,\mu m$.

In the one-moment scheme, the largest snow hydrometeors are little bit smaller smaller than in the two-moment scheme
and the distribution is not as steep for large particles. But the possibly most important difference of the size distributions from McFarquhar and Heymsfield (1997) and Seifert and Beheng (2006) stands out in particular: The number densities for small frozen hydrometeors in the two-moment scheme is orders of magnitude smaller than the one for the one-moment scheme. In this example, this is mainly due to the fact that processes creating small ice particles in the two-moment scheme are missing (pers. comm. Axel Seifert, 2016).
Altogether the differences in the size distributions of the different approaches have a large impact on the radiative transfer calculations because the hydrometeor size strongly influences the scattering properties of the particles. Nevertheless the size distributions lie in a realistic equivalent diameter range compared to measurements. Also, aircraft measurements have been criticised for having too many small particles due to shattering (e.g. Heymsfield, 2007) and the exact amount of smaller particles remains uncertain. However, for the millimetre and sub-millimetre range this is not critical because the sensitivity to particles
smaller than $100\,\mu m$ is small in this range (Eriksson et al., 2008). For the benefit of a second prognostic moment, namely the number density, and the potential of having both to also retrieve microphysical parameters we stay with the two-moment scheme.

It should be noted that beside the differences in the size distribution also the parameterisation of the particle shape is a crucial ingredient for radiative transfer modeling (e.g. Eriksson et al., 2015). However, atmospheric models normally have no detailed
information about the particle shape. A matching of the atmospheric model and the radiative transfer model with regard to this quantity would require more sophisticated assumptions in the atmospheric model.

## 3.2 ARTS

The hydrometeor size distributions of the particles have been implemented in a discretised form into ARTS using the same distribution function as the two-moment scheme from Seifert and Beheng (2006). As a second variable representing the micro-
physical characteristics of the hydrometeors we chose the particle mean mass $\bar{m}$ and calculate it by dividing the mass mixing ratio $M$ from ICON by the number mixing ratio $N$ from ICON. This has the advantage that the mass density Jacobians for a fixed particle mean mass (as opposed to a fixed particle number mixing ratio) correspond to the ones one would get from a one-moment bulk scheme. However, in the remainder of the article we will focus on the mass densities and only include the values for the mean particle masses $\bar{m}$ in the final results for the information content to see if the channels higher than $183\,GHz$
(see Sect. 5.1 for chosen set of channels) have a potential to measure cloud microphysical parameters such as hydrometeor size.

The scattering properties for the different hydrometeor types are defined as in Geer and Baordo (2014). For LWC and RWC we assume spherical particles, for IWC soft spheres with a density of $900\,kg\,m^{-3}$. For spherical particles we calculated the single scattering properties with Mie theory, using the program by Mätzler (2002).





For SWC, the scattering properties were taken from the database of Liu (2008) assuming sector-like snowflakes for channels up to and including 334 GHz and the data base of Hong et al. (2009) assuming aggregates for channels higher than that. Since the original Hong et al. (2009) database assumes a constant effective density for the aggregates and also is based on the old Warren (1984) refractive index we use a corrected version of the database, in which the absorption is rescaled using the Mätzler

(2006) parameterisation for the refractive index of ice. Rescaling is done by multiplication with $\mathrm{imag}(n)/\mathrm{imag}(n_0)$, where $n$ and $n_0$ are the refractive indices from Warren (1984) and Mätzler (2006), respectively. We apply the rescaling to obtain data for 183, 213, 243 and 266 K. The scattering extinction and the phase matrix remain unchanged, which means that the rescaling only applies to the absorption (see Brath et al. (in review, 2017) for details).

We use the Discrete Ordinate ITerative (DOIT, (Emde et al., 2004)) method to calculate the scattering within ARTS. We

calculate the Planck brightness temperatures for all side bands within our set of channels (see Sect. 5.1). We do not use an explicit sensor response function but perform monochromatic radiative transfer simulations for the center frequencies of the side bands in each channel and use the mean of the two brightness temperatures. We use a pencil beam with an incidence angle of $65°$ at the ground. For gas absorption we use the HITRAN (HIgh-resolution TRANsmission molecular absorption, Rothman et al. (2013)) database, the MT_CKD model (Mlawer et al., 2012) version 2.52 for the continuum absorption of water vapour

and the MT_CKD model version 1.00 for the continuum absorption of oxygen.

## 4 Reduction of degrees of freedom

In principle, an information content analysis quantifies the information that is contained in a measurement with a certain set of channels. The information leads to the reduction of the a priori error - the larger the information content, the larger the reduction. A quantification of the information is for example possible through calculating the reduction of the degrees of

freedom ($\Delta$DOF) for the analysis compared to the a priori state, or through calculating the entropy $S$ of the two states (e.g. Rodgers, 2000; Di Michele and Bauer, 2006). In this study we use the reduction of the degrees of freedom $\Delta$DOF, which is defined by

$$\Delta\mathrm{DOF} = \mathrm{trace}\left(I - S_r S_a^{-1}\right), \tag{2}$$

with the unity matrix $I$, the a priori covariance matrix $S_a$ and the a posteriori, or analysis error covariance matrix $S_r$. $S_r$ is

defined according to the optimal estimation method as the reciprocal sum of the a priori and measurement error $S_y$:

$$S_r = \left(S_a^{-1} + J^T S_y^{-1} J\right)^{-1}. \tag{3}$$

$S_y$ is transformed from measurement space into state space with the transpose of the Jacobian $J$. We set the measurement error to 1 K for each channel and assume that it is uncorrelated between channels, therefore $S_y$ is a diagonal matrix with $1\,\mathrm{K}^2$ on the diagonal. Our assumption on $S_a$ is described further below in Sect. 4.2.

If the analysis error after the measurement is equally large as the a priori error before, $\Delta$DOF is zero and no information was gained. The closer the analysis error is to zero, the larger is $\Delta$DOF, with a maximum (in reality unreachable) value equal to the number of channels, in our case 24. $\Delta$DOF can also be interpreted as pieces of information. If we have 1 piece of information





we can retrieve 1 quantity, for example the hydrometeor path, i.e., the column integrated mass of the hydrometeor. If we have two pieces we can get two quantities, for example the hydrometeor mass in two different heights.

For our analysis, we need the portion of the information content which is associated with the specific hydrometeors. The method we chose is a linear splitting of the trace in the definition of $\Delta$DOF to the block matrices which correspond to the

respective quantity (H2O, IWC, LWC, SWC, RWC). However, we would like to stress that this is an approximation and does not consider the cross-correlations between the various hydrometeors.

## 4.1 Jacobians

The calculation of the Jacobians $J$ generally requires the following steps: (1) calculate the brightness temperature $T_B^c$ for a specific channel $c$ for the unperturbed atmosphere, (2) perturb one atmospheric quantity $x$, for example IWC, by a perturbation $\delta$

and again simulate the perturbed brightness temperature $[T_B^c]_\delta$ for that channel, (3) divide the difference of the two brightness temperatures by the perturbation. If, as in our analysis, height resolved Jacobians are required, the perturbation has to be applied successively to each of the height levels $k$. Note that a perturbation at a distinct height level $k$ strictly spoken means a perturbation of the respective quantity at the two adjacent height layers which the radiation passes.

The Jacobian $J_{c,k}$ at height $k$ and for a channel $c$ is therefore given by:

$$J_{c,k} = \frac{[T_B^c]_{k,\delta} - T_B^c}{\delta},\qquad(4)$$

with $[T_B^c]_{k,\delta}$ as the simulated brightness temperature if we perturb the quantity $x_k$, which denotes the value of $x$ at the height level $k$.

For our analysis, we define $\delta$ as a relative perturbation of $x_k$, as opposed to using an absolute value that is independent of the specific value of the $x_k$. This is especially useful for the calculation of Jacobians for the hydrometeor profiles. First, the values

of $x$ over height span several orders of magnitude. The use of a relative perturbation ensures that the perturbation is always small compared to the original value, and linearity can be assumed. Second, the hydrometeor profiles are discontinuous and do not exist at all heights. Using the relative perturbation ensures that we only perturb the profile where hydrometeors exist in the first place. In our analysis, we use $\delta = 1\%$ for each quantity (including water vapour, which in the following is referred to as H2O) and at all heights.

Relative Jacobians also correspond to the retrieval of a quantity in logarithmic space. Regarding $1 + \delta$ as the development of the natural logarithm for small $\delta$ we can find that $\delta = \ln(x_{k,\delta}) - \ln(x_k) = \Delta \ln x_k$, with $x_k$ as unperturbed value at height level $k$ and $x_{k,\delta}$ as perturbed value. The Jacobian then is

$$J_{c,k} = \frac{[T_B^c]_{k,\delta} - T_B^c}{\Delta \ln(x_k)},\qquad(5)$$

As stated above, this corresponds to a retrieval in natural logarithm space. In the remainder of the article, we will entirely stay

in the framework of a logarithmic retrieval.

We calculate the Jacobians for each of the two sidebands (see Sect. 5 for the definition of channels and side bands) and use the mean of the two Jacobians for the subsequent analysis. We use the same height grid in ARTS as in the ICON simulation.





We calculated the Jacobians for the H2O volume mixing ratio (VMR), the hydrometeor mass densities $M$ and the hydrometeor mean mass $\bar{m}$. For the analysis, we use the Jacobians in units of K/100% as calculated by ARTS. For the purpose of showing them in the following sections, they are normalised by the height layer thickness. Note that the height layers broaden with increasing height. This yields the unit K/(%·km), which appears in the plots. Thus we ensure the comparability of the Jacobians

at different height levels. All calculations, however, have been performed on the unnormalised values.

Note that Eq. (4) and (5) only conceptually describe the Jacobian calculation. In practice, we do not make a fully independent $T_B$ calculation for each perturbation, since this is computationally very inefficient for the iterative scattering solver used (Emde et al., 2004). Instead, the scattering solver for the perturbations gets the reference result as a first guess, which saves most of the iterations that would otherwise be needed.

## 4.2  A priori covariance

The final ingredient to calculate the information content of a measurement is the a priori covariance error matrix $S_a$. In our ICON model framework we have the opportunity to calculate that matrix directly out of the model data as the covariance of the different quantities on different height levels. This means that we take the model mean state as a priori state, and the full variability of the model on the chosen domain (state domain) as its uncertainty. We have to keep in mind that the simulation

case is a spring day in the mid-latitudes and that the variability is therefore limited. In reality, the inter-seasonal and also the inter-annual variability of the atmospheric conditions in the mid-latitudes would be even larger than what we get from our spring case. To cover the broadest possible statistics from this case, we use the entire three days of the model simulation. This especially means that we cover clear sky cases as well as frontal cases with our a priori assumption. We find that already in this limited case, the variability of the hydrometeors is very large (up to 670 in ln space). On the contrary, the variability of H2O

is quite low (order of 1) because it corresponds more to the overall synoptic situation, which only changes on time scales of days. Also, the dynamical range of H2O is smaller than the one for the hydrometeors. Note that since we use relative Jacobians (see Sect. 4.1), i.e. a retrieval in natural logarithm space, we also need to calculate the covariance in ln-space. To enable this calculation, zero values were treated by setting them to the smallest possible float difference (2.22e-16 in the respective unit).

Figure 2 shows the a priori covariance in ln space and the corresponding correlation matrix defined as $C_{i,j} = S_{i,j}/\sqrt{S_{ii}S_{jj}}$.

The 25 block matrices give the covariance and correlation of pairs of model variables on their 49 height levels. Note that we have to skip the uppermost 50th height level from ICON because ARTS requires one level on top of the "cloud box" which defines the cloudy area where scattering is calculated. Since the matrices are symmetric, only the lower triangle is shown for clarity. The covariance naturally is largest at the height levels where hydrometeors reside and goes up to 670 in units of the natural logarithm on the diagonal of the SWC×SWC block matrix. The covariance for LWC with itself is comparably small.

The one for H2O is smooth and the spread of values is very small compared to the one for hydrometeors (order of 1). This reflects the much lower variability and smaller dynamical range of H2O compared to hydrometeors. Note that the regarded scene is a short period spring time front in the mid-latitudes. In winter, the correlations likely will look different, when snow for example reaches the ground.




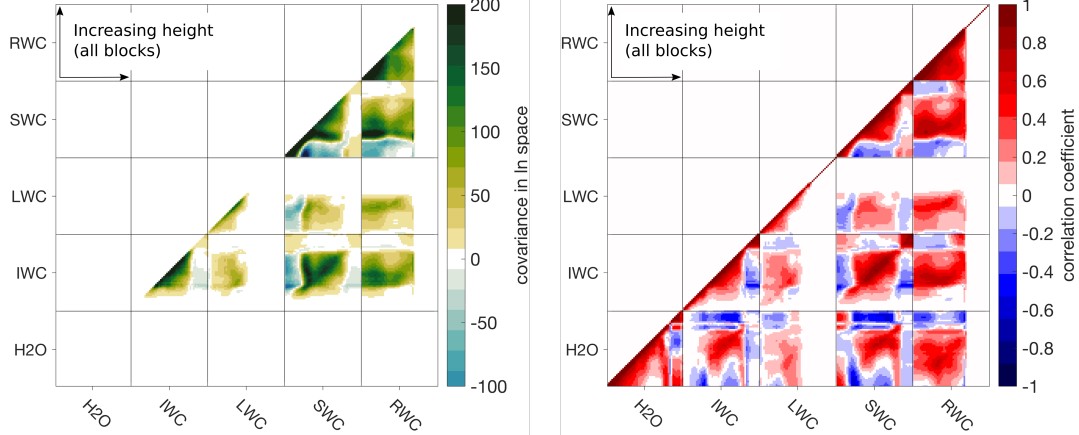

**Figure 2.** A priori error covariance (left) and the corresponding correlation matrix (right). Only the lower triangle is shown for clarity, since the matrices are symmetric. The block matrices correspond to the (auto-)correlation for one or between two quantities. They have the dimension of 49×49 height levels each, the height increases within the blocks from left to right and from bottom to top. Note that the variability of $H_2O$ is so small in comparison to the hydrometeors (in the range 0 to 1) that it cannot be seen in the a priori covariance (left).

There is a rich structure of autocorrelations and correlations between different hydrometeors and over different height levels. For example, the autocorrelation for SWC is positive in the upper layers but negative for a combination of upper and lower levels. This is also seen in the correlation of SWC and RWC, which are anti-correlated at lower levels. The autocorrelation for RWC is entirely positive below the melting layer, and zero above, where rain does not exist. This is due to the prognostic rain falling down through the different height levels over time. This means, in order to have precipitation in lower levels it needs to be present in the upper levels in the first place in order to fall down.

In the correlation plot also the structures for $H_2O$ become evident. It is positively correlated with hydrometeors within the clouds and precipitation regions, since the atmosphere is near saturation there. The negative correlation at lower regions may be due to the fact that in case hydrometeors are present in sub-saturated regions evaporation takes place which decreases the hydrometeor mass but increases the $H_2O$ mass. At higher regions above the clouds it may be spurious and stem from the fact that there are only numerical artefacts of very small amounts of hydrometeors in comparison to realistic amounts of $H_2O$.

The covariance and correlations stem from the ICON model simulation, i.e., from about 300 000 near-realistic profiles. With the ICON matrix we automatically get the cross-correlations for the different hydrometeor types and do not have to add additional assumptions. However, one has to be aware that there are model inherent correlations due to the microphysical parameterisation, which can cause some artefacts. Also, the terrain following coordinates of ICON cause slightly larger covariances in the lower levels for $H_2O$ and rain, which both are present near the ground. Idealised covariance matrices might be constructed instead, but they have different flaws as well and contain many more or less arbitrary assumptions, especially for the cross-correlation of hydrometeors. We therefore chose the model based a priori covariance matrix to perform our study, keeping in mind the downside that the model introduces some artificial correlations between the hydrometeors.



## 5   Setup

### 5.1   Channels

In our analysis, we use radiometer channels as applied on the International Sub-millimeter Microwave Airborne Radiometer (ISMAR, Fox et al. (2017)) and on the Microwave Airborne Radiometer Scanning System (MARSS, McGrath and Hewison

(2001)). Both were employed in a recent flight campaign (Brath et al., in review, 2017) and cover channels from 89.0 GHz up to 874.4 GHz. They include the AMSU-B channels and the ICI setup, with the exception that the ICI-1, ICI-2 and ICI-3 channels have a slightly different distance from the $H_2O$ absorption peak at 183.31 GHz (Pica et al. (2012)). We will later use the three 183.31 GHz channels from MARSS instead. We further complement the setup by two channels at 23.8 and 50.1 GHz from the Dual-frequency Extension to In-flight Microwave Observing System (Deimos, Hewison (1995)) to account for lower

frequency precipitation channels, which are also part of MWI. The resulting set of channels, their side bands, and the respective instrument they belong to is given in Table 2.

With this setup it is possible to investigate the principle interdependencies of the information content on different hydrometeors from a set of channels, which is capable to observe liquid and frozen cloud as well as precipitation hydrometeors. But it is also possible to put a special focus on the upcoming ICI instrument on MetOp-SG, which employs channels from 183 GHz

upwards to gain more detailed information about cloud ice, its microphysical properties, and maybe even some more profile information than the instruments that are currently employed in the different satellite missions.

### 5.2   Atmospheric profiles

To facilitate the analysis we have calculated a mean profile (Fig. 3) from 10 000 ICON profiles, which each are amongst the extremes for one specific hydrometeor or the humidity. To chose them, we calculated the hydrometeor paths for each

hydrometeor and each atmospheric column. To exclude outliers we disregard the columns with a path larger than the 99th percentile. For the rest of the columns we chose 10 000/7 largest paths for H2O, LWC, IWC, RWC, SWC, hail and graupel (the "divided by seven" stems from the 7 quantities we loop over). This ensures that a considerable amount of each hydrometeor, except for hail and graupel, which do only consist in very small amounts over the whole simulation, is contained in the profile. However, since the 10 000 atmospheres are not required to contain all hydrometeor types at once, this gives us 10 000 cloudy

profiles and on average a mean profile which is not extreme for any hydrometeor and which is comparably smooth. This mean profile that follows from these choices contains realistic amounts of hydrometeor masses and cloud and precipitation are located in physically reasonable height ranges. It has to be kept in mind, though, that this may lead to an unlikely combination of hydrometeors, such as LWC and SWC being present in the same atmospheric column. Therefore, in Sect. 6.4, we will also show results from a set of 90 individual cloudy atmospheric columns drawn directly from the selected ICON simulation to

consolidate the results from the idealised atmosphere.

We use this mean profile as a "base profile" containing all hydrometeors. From this base profile we create atmospheres with different combinations of cloud hydrometeors by taking out or putting in specific hydrometeors. A similar approach has been used by Guerbette et al. (2016), who progressively put in cloud hydrometeors to quantify their respective influence on





**Table 2.** Selected set of channels from the instruments MARSS, ISMAR and Deimos. Channels equal or similar to the ones of the MetOp-SG mission (ICI and MWI) are marked in the right column.

| Center frequency [GHz] | Side bands [GHz] | Bandwidths [GHz] | Instrument | METOP-SG |
|---|---|---|---|---|
| 23.8 | ±0.07 | 0.127 | Deimos | MWI-2 |
| 50.1 | ±0.08 | 0.082 | Deimos | near MWI-4 |
| 89.0 | ±1.1 | 0.65 | MARSS | MWI-8 |
| 118.75 | ±1.1 | 0.4 | ISMAR | near MWI-12 |
| 118.75 | ±1.5 | 0.4 | ISMAR | near MWI-11 |
| 118.75 | ±2.1 | 0.8 | ISMAR | near MWI-10 |
| 118.75 | ±3.0 | 1.0 | ISMAR | near MWI-9 |
| 118.75 | ±5.0 | 2.0 | ISMAR | |
| 157.05 | ±2.6 | 2.6 | MARSS | |
| 183.31 | ±1.0 | 0.45 | MARSS | near ICI-3 |
| 183.31 | ±3.0 | 1.0 | MARSS | near MWI-17, near ICI-2 |
| 183.31 | ±7.0 | 2.0 | MARSS | near ICI-1 |
| 243.20 | ±2.5 | 3.0 | ISMAR | near MWI-18, ICI-4 |
| 325.15 | ±1.5 | 1.6 | ISMAR | ICI-7 |
| 325.15 | ±3.5 | 2.4 | ISMAR | ICI-6 |
| 325.15 | ±9.5 | 3.0 | ISMAR | ICI-5 |
| 424.70 | ±1.0 | 0.4 | ISMAR | |
| 424.70 | ±1.5 | 0.6 | ISMAR | |
| 424.70 | ±4.0 | 1.0 | ISMAR | |
| 448.0 | ±1.4 | 1.2 | ISMAR | ICI-10 |
| 448.0 | ±3.0 | 2.0 | ISMAR | ICI-9 |
| 448.0 | ±7.2 | 3.0 | ISMAR | ICI-8 |
| 664.0 | ±4.2 | 3.0 | ISMAR | ICI-11 |
| 874.4 | ±6.0 | 3.0 | ISMAR | |

the brightness temperature in the 183 GHz channel of the humidity sounder SAPHIR on Megha-Tropiques. In our study, we investigate if the information about one hydrometeor type depends on the presence of another. For example, we can have an atmosphere which contains only LWC or only IWC, or one which contains the two hydrometeor types IWC and RWC. All in all 16 combinations including the clear sky are possible. We are aware that not all combinations are physically possible and realistic, such as an atmosphere only containing SWC, but we want to understand how the measurement of one hydrometeor is in principle influenced by the others. Therefore we regard all mathematically possible combinations.





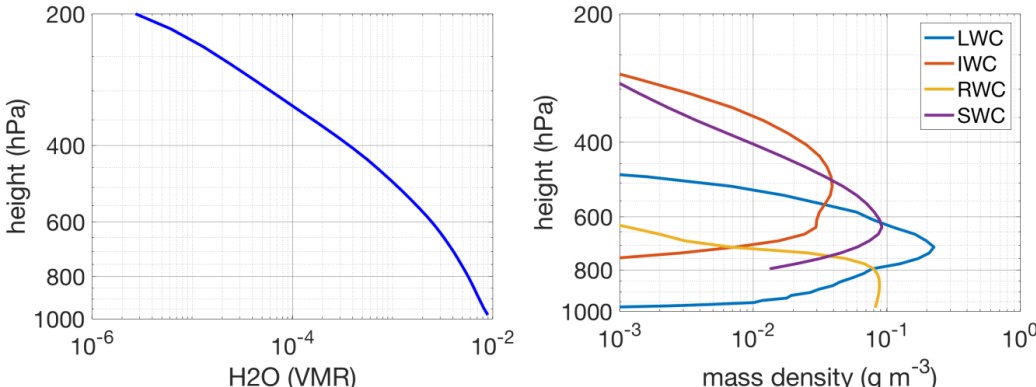

**Figure 3.** Idealised atmospheric base profile. H2O volume mixing ratio (VMR, left) and particle mass densities (right) for LWC, IWC, RWC and SWC.

We denote the different atmospheres by an $a_X$, where $X$ contains the composition of the atmosphere. LWC is denoted by $L$, IWC by $I$, RWC by $R$ and SWC by $S$. For example, the atmosphere containing IWC and LWC is called $a_{IL}$. The clear sky case ("Vapour") is called $a_V$. Note that H2O is present in all of the atmospheres, even though it is not explicitly included in $X$ in case hydrometeors are present. An overview over the atmospheric compositions is given in Table 3.

## 6 Results

### 6.1 Brightness temperature spectra

The brightness temperatures for the different atmospheric compositions are shown in Fig. 4 for an emissivity of $\epsilon = 0.6$. Naturally, the brightness temperature spectra differ for the different atmospheric compositions, and they differ by up to 80 K, depending on the measurement channel. In the window channels, the spread in the spectra is particularly large, while in the sounding channels near the absorption peaks the spread is smaller. For channels close to absorption line centres the spectra

**Table 3.** Atmospheric compositions used in the analysis of the dependency of the information content on the atmospheric composition.

| | | $a_V$ | $a_L$ | $a_R$ | $a_I$ | $a_S$ | $a_{LR}$ | $a_{LI}$ | $a_{LS}$ | $a_{RI}$ | $a_{RS}$ | $a_{IS}$ | $a_{LRI}$ | $a_{LRS}$ | $a_{LIS}$ | $a_{RIS}$ | $a_{LRIS}$ |
|---|---|---|---|---|---|---|---|---|---|---|---|---|---|---|---|---|---|
| | | | | | | | | | | | | | | | | **Atmosphere** | |
| Hydrometeors | Vapour | X | X | X | X | X | X | X | X | X | X | X | X | X | X | X | X |
| | LWC | | X | | | | X | X | X | | | | X | X | X | | X |
| | RWC | | | X | | | X | | | X | X | | X | X | | X | X |
| | IWC | | | | X | | | X | | X | | X | X | | X | X | X |
| | SWC | | | | | X | | | X | | X | X | | X | X | X | X |





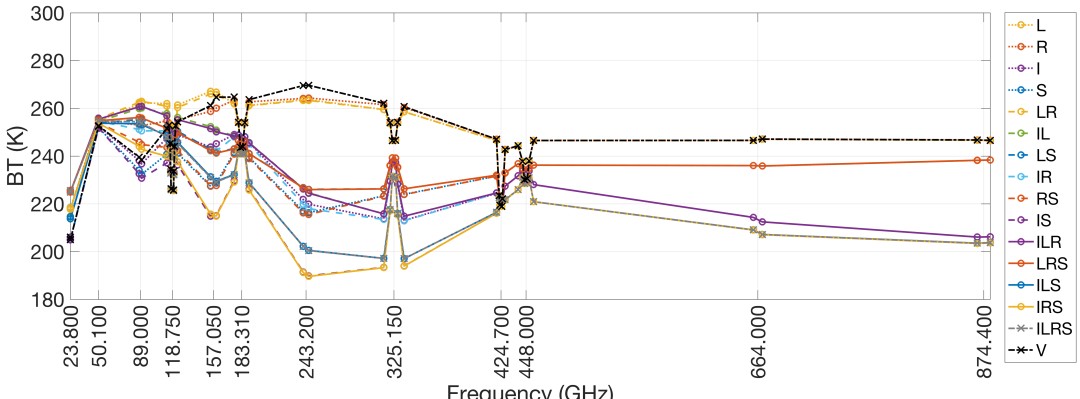

**Figure 4.** Brightness temperature spectrum for $\epsilon = 0.6$ for the 16 combinations of the base profile. The legend corresponds to the composition suffixes $X$ defined in Table 3. The labels on the abscissa are the center frequencies of the channels. The curve labeled "V" is for H2O only, without any hydrometeors. Although not mentioned in the legend, vapour is also present in all the other calculations.

almost lie on top of each other because here the atmosphere is opaque due to H2O absorption. Some of the different composi-
tions such as $a_L$ and $a_{LR}$ or $a_{IS}$ and $a_{IRS}$, have almost the same spectra except for differences in the window channels below
118.75 GHz, which implies that some hydrometeors are invisible to channels higher than that, e.g. due to the high opacity of
the H2O, which shields hydrometeors in lower atmospheric levels or due to a general insensitivity of the channel to the respec-
tive hydrometeor type. Hydrometeors can also be shielded by other hydrometeors, that create a radiative background through
absorption or scattering which is similar to the radiative signal of the hydrometeor in question, which masks the signal. In the
following we investigate this further. We will have a closer look at the sensitivities of the brightness temperature to changes in
the hydrometeor mass, namely the Jacobians $J$, in the absence and presence of other hydrometeors.

## 6.2   Cloudy Sky Jacobians

We first look at the Jacobians for H2O for the clear sky case $a_V$ and the all-hydrometeors case $a_{ILRS}$ (Fig. 5). H2O has the
advantage that its profile is smooth and continuous, contrary to the hydrometeor Jacobians which per definition of the relative
perturbation only exist where the cloud hydrometeors reside and which decrease to zero at the cloud edge with a steep gradient.
With the chosen surface emissivity of $\epsilon = 0.6$, for $a_V$, H2O gives a warming signal from the lower atmosphere ($> 500$ hPa)
at channels from 157.05 GHz downward and at 243.2 GHz. For channels from 183.31 GHz upward it gives a small cooling
signal at higher levels ($< 650$ hPa). Hereby "warming signal" ("cooling signal") in our context means that an increase of the
amount of vapour or hydrometeor content leads to a warming (cooling) of the resulting brightness temperature at the top of the
atmosphere. This is mainly due to the fact that the Jacobians for the sounding channels higher than 183 GHz peak higher up in
the atmosphere than the Jacobians of the lower channels and that these higher regions are very cold compared to the ground.





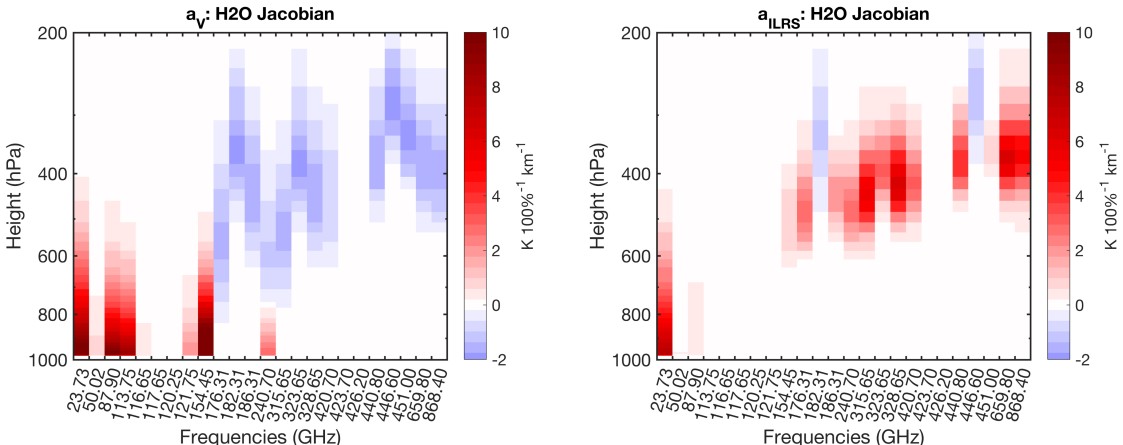

**Figure 5.** H2O Jacobians for the clear sky case $a_V$ (left) and for the all-hydrometeor case $a_{ILRS}$ (right) for an emissivity $\epsilon = 0.6$. Averages of the sidebands are shown, the labels on the abscissa denote the left sideband of the channel.

This picture changes dramatically in the presence of all considered cloud hydrometeors ($a_{ILRS}$). Except for the most central ones of the sounding channels at 183.31 GHz and 448.0 GHz the H2O signal in this case is entirely positive. The positive signal in between the channels at 157.05 GHz and 23.8 GHz decreases to almost zero. The sensitivity of the measured brightness temperature to changes in the H2O is highly dependent on the presence of clouds.

The H2O example illustrates the general principle of these interactions well: If the radiative background is cold, then the presence of a scattering or absorbing species tends to increase the brightness temperature. Conversely, if the background is warm, then the species tend to reduce the brightness temperature. For H2O at high frequencies, the presence of frozen hydrometeors in the upper troposphere, which have a cooling signal due to scattering, turns the scene from a warm background case to a cold background case.

Figure 6 illustrates Jacobians from atmospheres with one single liquid hydrometeor type each, i.e., $a_L$ and $a_R$ for both, LWC or RWC, along with the corresponding H2O Jacobians. Because we used a relative perturbation for the calculation of the Jacobians (Eq. (5)), the cloud Jacobians naturally only exist at those heights where LWC or RWC exist (cp. Fig. 3). These heights are indicated in the figures for two different thresholds for the respective mass densities. Mainly the channels below 325.15 GHz (LWC) respectively 243.2 GHz (RWC) are sensitive to the liquid hydrometeors. The window channels at

23.8 GHz, 50.1 GHz and 89.0 GHz give a warming signal at all heights, the outermost channel at 118.75 GHz (and 157.05 GHz for RWC) have a warming signal at lower levels and a cooling signal at upper levels. Note that a higher surface emissivity, i.e., a radiatively warmer surface, reduces the warming signal from LWC and RWC in the window channels, since the surface provides a warmer background in that case (not shown). The Jacobians for H2O change in the presence of LWC or RWC. Apart from 23.8 GHz the warming in the lower channels is considerably reduced. The warming signal from H2O alone is

partly shielded by the warming signal from LWC and RWC.





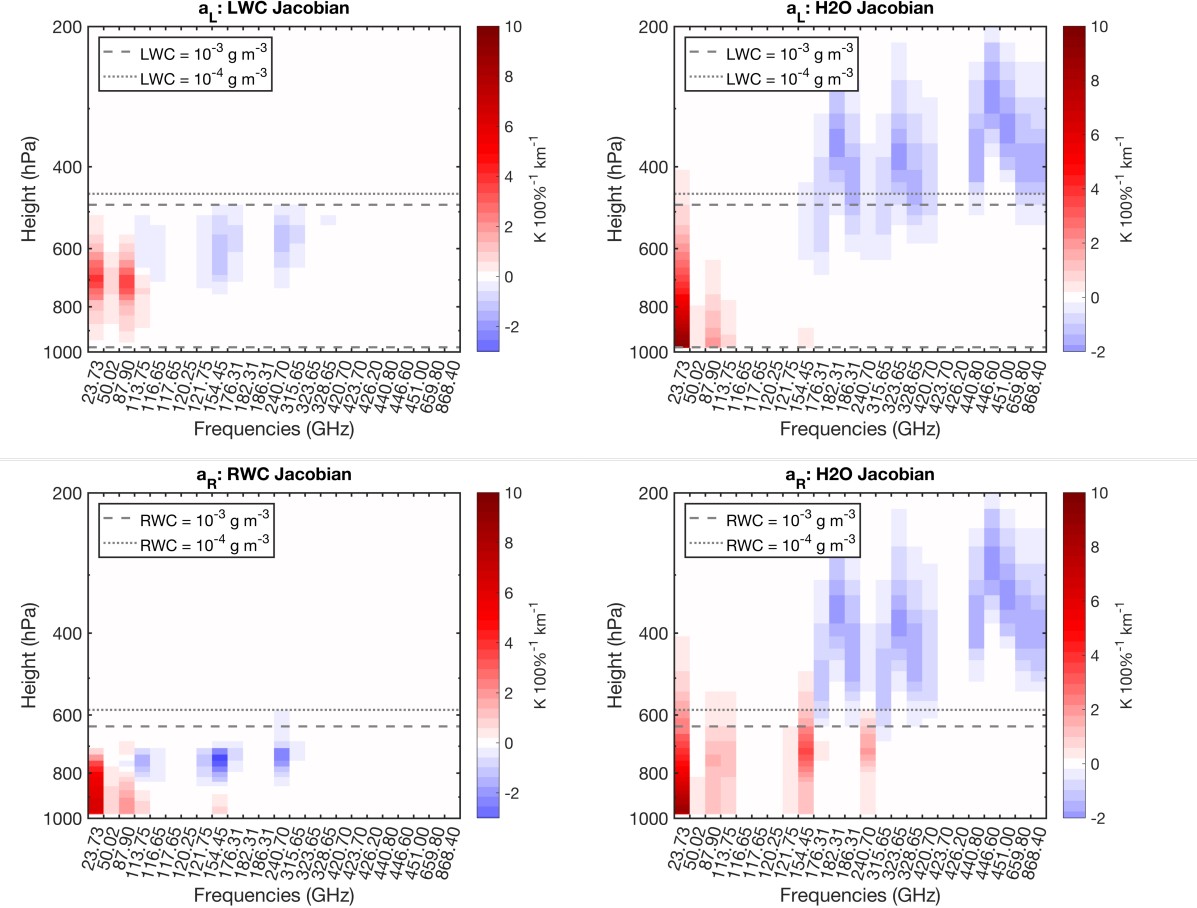

**Figure 6.** LWC (top left) and H2O (top right) Jacobians for $a_L$, RWC (bottom left) and H2O (bottom right) Jacobians for $a_R$ ($\epsilon = 0.6$). The dashed (dotted) grey line denotes the height in which the mass content of the respective hydrometeor is nearest $10^{-3}$ g/m$^3$ ($10^{-4}$ g/m$^3$). Note that both hydrometeor types reach far down to the ground such that the lower edges are not always visible in the plots.

The frozen hydrometeor types IWC and SWC generally give a cooling signal (Fig. 7) since they mainly act as scatterers rather than absorbers in the selected channel range. Also, they exist at low ambient temperatures, and even their emission would cause a cooling signal. The upper channels above 157.05 GHz are sensitive to these hydrometeor types. For SWC a considerable signal also comes from the channels at 50.1, 89.0 and the outermost 118.75 GHz channel. The highest channels at

5   664.0 and 874.4 GHz are more sensitive to IWC than to SWC because the scattering efficiency in these two channels is larger for the smaller ice hydrometeors than for the larger snow hydrometeors.

The corresponding H2O Jacobians are considerably changed at channels above 157.05 GHz: The cooling signal from the clear sky H2O Jacobians (Fig. 5) is turned into a warming one except for the sounding channels closest to the absorption



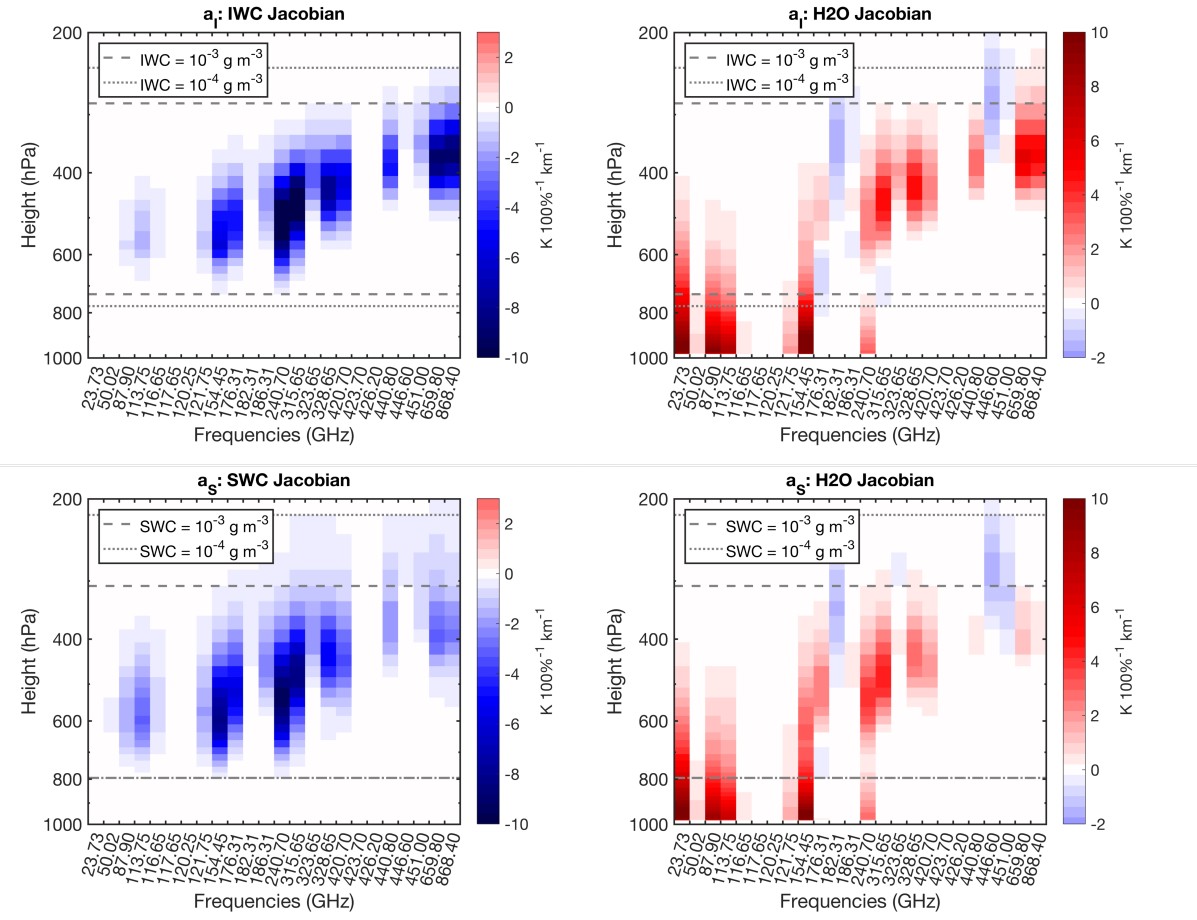

**Figure 7.** Same as Fig. 6 but for IWC (top) and SWC (bottom).

lines. This is in accordance with the findings of Guerbette et al. (2016) who found such a change of sign in the lowest-peaking SAPHIR channels near 183 GHz in the presence of high concentrations of snow above 500 hPa.

Next, we will go deeper into the interdependencies of the hydrometeor Jacobians. For this purpose, we reduce the view chosen in the previous figures, and only show the Jacobians for the channels at 89.0 GHz and 243.2 GHz as line plots (Figs. 8

5   and 9). In these channels we expect to get a signal from all cloud hydrometeors, although 89.0 GHz is more sensitive to the liquid hydrometeors and 243.2 GHz is more sensitive to the frozen hydrometeors. We show the Jacobians for each cloud hydrometeor type for the cases where only that specific type is present, one other hydrometeor type is present, or all types are present in a combined plot. At the example of cloud ice these are IWC Jacobians for the cases $a_I$, $a_{IL}$, $a_{IS}$, $a_{IR}$, and $a_{ILRS}$.

For the LWC Jacobians (Fig. 8, top row), the lines group in two sets in both channels. In the 89.0 GHz channel, the signal

10   is reduced in the presence of RWC in the lower levels. The presence of frozen hydrometeors does not alter the signal much. The pairs not including RWC, $a_{IL}$ and $a_{LS}$, almost give the same Jacobian as $a_L$, while both, $a_{LR}$ and the all-hydrometeors





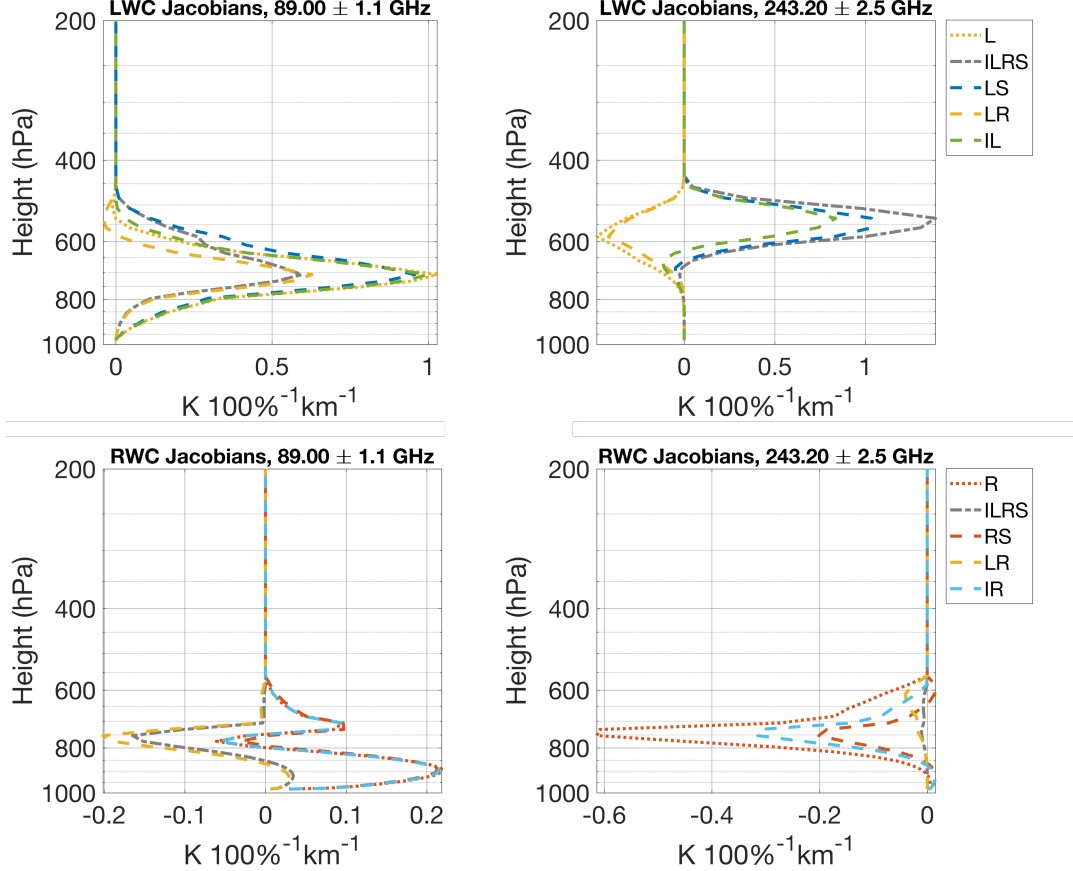

**Figure 8.** LWC (top row) and RWC (bottom row) Jacobians for the 89.0 GHz (left column) and the 243.2 GHz (right column) channel ($\epsilon = 0.6$). Shown are atmospheres containing pairs of hydrometeors and the all hydrometeor case $a_{ILRS}$. The labels in the legend correspond to the atmospheric composition suffix $X$. Note different values on the abcissa in the different plots.

case $a_{ILRS}$ have a smaller peak and are very close up to about 700 hPa. Above that level, SWC has a greater influence. The Jacobian for $a_{LS}$ deviates from the one of $a_L$ and the all-hydrometeor case $a_{ILRS}$ approaches the curve for the case $a_{LS}$. The change of behaviour near 700 hPa is due to the height ranges where SWC, or RWC respectively, occur. Near this height, the melting layer of the idealised atmosphere is located (see Fig. 3).

5    The 243.2 GHz channel has its largest sensitivity for the detection of RWC higher up in the atmosphere than the channel at 89 GHz and therefore does not exhibit such a transition. The two cases $a_L$ and $a_{LR}$ which only contain liquid hydrometeors have negative LWC Jacobians, while the presence of any frozen hydrometeors results in positive LWC Jacobians. The largest signal comes from the all-hydrometeors case $a_{ILRS}$, the smallest positive one from the combination of LWC with IWC, i.e. $a_{IL}$.





This again can be understood if we think of the other cloud hydrometeors as contributors to the mixture of signals from all heights and hydrometeors, which result in the respective brightness temperature. The other cloud hydrometeor types, the surface and the H2O create a radiative background for the hydrometeor type in question. At 89.0 GHz, the presence of RWC already increases the brightness temperature, therefore the emission from LWC only adds a smaller positive increment compared to an

atmosphere where only LWC is present. At 243.2 GHz, the scattering by frozen particles decreases the measured brightness temperature such that the emission by LWC adds a positive increment instead of a negative one if no IWC or SWC is present. These effects are not linear and can not just be added up.

For RWC (Fig. 8, bottom row), in the 89.0 GHz channel we also find a grouping of the Jacobians similar as for LWC. For RWC the sign of the signal depends on the height. The lower levels cause a warmer background, such that the higher levels'

contribution is negative compared to it. In the 243.2 GHz channel, the signal from rain is negative with the exception of a small positive contribution near the ground. The addition of any of the other hydrometeor types decreases the amplitude of the Jacobian. Each hydrometeor type alone, LWC, IWC and SWC, gives a cooling signal and therefore causes a colder background in the mixture compared to the case where only RWC is present.

Figure 9 shows the corresponding figures for the frozen hydrometeors IWC and SWC for the two channels. Since here the

main interaction of the particles with the radiation is scattering, the signal is robustly negative. In the 89.0 GHz channel, the IWC Jacobians for $a_I$ and $a_{IS}$ as well as the SWC Jacobians for $a_S$ and $a_{IS}$ are very close. IWC and SWC only little influence each other in that channel. The addition of liquid hydrometeors below the frozen ones leads to a stronger signal for IWC and SWC, because the liquid hydrometeors provide a warmer background for the frozen hydrometeors. In the 243.2 GHz channel, the picture is almost the same. In this channel, however, the signals from the frozen hydrometeors are much stronger, and the

combination of IWC and SWC results in a considerably stronger cooling signal for both cloud hydrometeor types. Therefore the Jacobians for the all-hydrometeors case $a_{ILRS}$ lie between $a_{IS}$ and the other shown cases.

### 6.3 Information content

The question is how much the information we gain from the observation depends on the composition of the atmosphere. Fig. 10 and Table 4 summarise $\Delta$DOF for the 16 different atmospheric compositions, observed with the full set of channels

and observed with the ICI channels. We did not show the analysis for particle mean mass $\bar{m}$ Jacobians in the previous sections, but we include the values for their information content in this section to show the potential of new sensors in this frequency range with regard to observing microphysical properties of the particles.

For the full set of channels, the total information content reaches up to values as high as 15.42 for $a_{ILRS}$ and is lowest for the clear sky case $a_V$ with 3.55 (Table 4).

Naturally, the more complex the atmosphere is, the higher is the overall possible total information content since the initial degrees of freedom for these cases are more numerous and a greater portion of the channels can be used to reduce them (Fig. 10). Also, it is very clear that once frozen hydrometeors are present in the atmosphere the information content considerably increases by more than 3 compared to atmospheres which only contain liquid hydrometeors, because the Jacobians for frozen atmospheres have larger values and also much more channels are sensitive to these hydrometeor types (see Figs. 6 and 7).



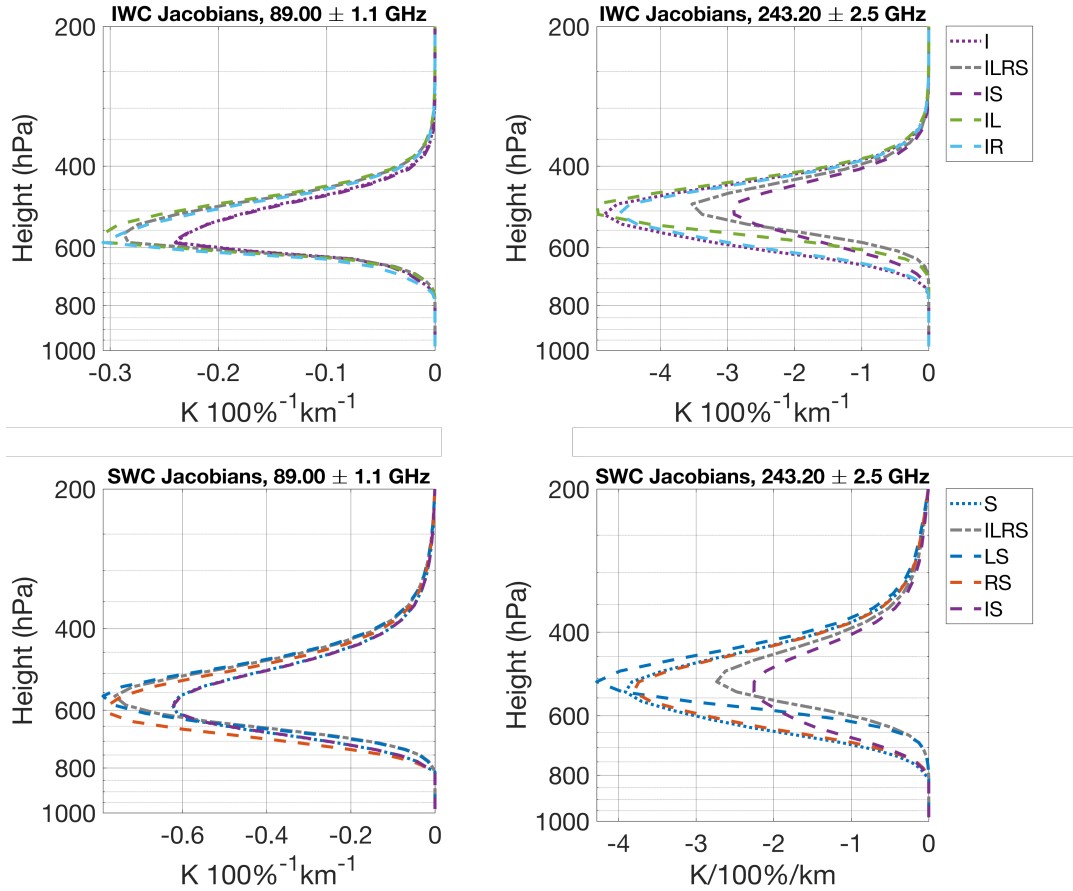

**Figure 9.** Same as Fig. 8, but for IWC (top row) and SWC (bottom row).

The presence of frozen hydrometeors slightly reduces the information about liquid ones. Also IWC and SWC compete for the information. If both are included, both their information contents slightly decrease by less than 1. This mirrors the behaviour of the Jacobians discussed above. If both frozen hydrometeor types are present, the absolute values of the Jacobians decrease. The overall picture is the same for both, land and ocean, with only slight changes in the ranking of the total information content

5   (not shown). For the higher land emissivities, the cases $a_{ILRS}$, $a_{IL}$ and $a_{LR}$ are shifted one position down each in that case, which means that $a_{IRS}$, $a_{RS}$ and $a_R$ are shifted one position up each. However, the respective pairs have very similar values in both cases and small changes in the information content easily lead to a slightly different ranking of the atmospheres.

IWC and SWC give on average the highest $\Delta$DOF of 4.99 and 4.84 (see Table 4). The spread is slightly higher for SWC, but the minimum information content is high for both, 4.28 for IWC and 3.89 for SWC. The mean information content of 2.36

10   for LWC is much lower, and we get least mean information about RWC (1.81). The finding for LWC is in agreement with the study by Crewell et al. (2009), who found that with a limited set of channels in the window regions at 31, 90 and 150 GHz the information content yields up to 2 pieces of information and a profile retrieval for LWC is not possible. Our channels larger than





**Table 4.** Information content $\Delta$DOF. Shown are mean, minimum and maximum of the total $\Delta$DOF and of the the $\Delta$DOFs for hydrometeor mass densities and the corresponding particle mean masses ($\bar{m}$) found in the set of 16 atmospheric combinations for all channels and for the ICI like channels.

| | all channels | ICI channels |
|---|---|---|
| | min \| **mean** \| max | min \| **mean** \| max |
| total | 3.55 \| **11.23** \| 15.42 | 2.73 \| **7.60**\| 9.99 |
| H2O | 0.66 \| **1.68** \| 3.55 | 0.50 \| **1.14** \| 2.73 |
| LWC | 2.05 \| **2.36** \| 2.93 | 0.69 \| **0.94** \| 1.62 |
| IWC | 4.28 \| **4.99** \| 5.63 | 3.80 \| **4.50** \| 5.43 |
| RWC | 1.14 \| **1.81** \| 2.94 | 0.02 \| **0.27** \| 1.41 |
| SWC | 3.89 \| **4.84** \| 5.75 | 3.16 \| **4.25** \| 5.49 |
| LWC $\bar{m}$ | 0.11e-3 \| **0.35e-3** \| 1.02e-3 | 1.1e-4 \| **3.5e-4** \| 0.1e-4 |
| IWC $\bar{m}$ | 1.25 \| **2.38** \| 3.49 | 0.69 \| **1.51** \| 2.44 |
| RWC $\bar{m}$ | 0.74 \| **1.01** \| 1.39 | 1.0e-4 \| **6.8e-2** \| 0.37 |
| SWC $\bar{m}$ | 1.28 \| **1.79** \| 2.60 | 0.96 \| **1.40** \| 1.92 |

157 GHz do not add much more information because the channels are only little sensitive to LWC (see Fig. 6). For H2O under clear sky conditions, we gain a maximum $\Delta$DOF of 3.55, which decreases in the presence of clouds. Once hydrometeors are present in the atmospheric column, the information content is considerably reduced down to 0.66 for the case $a_{IRS}$. For clear sky, with the large set of channels we expected better sounding capacities of the sensor (i.e., a larger $\Delta$DOF). The comparably low values we get are likely due to the high a priori autocorrelation of H2O which is inherent in the model (cp. Fig. 2).

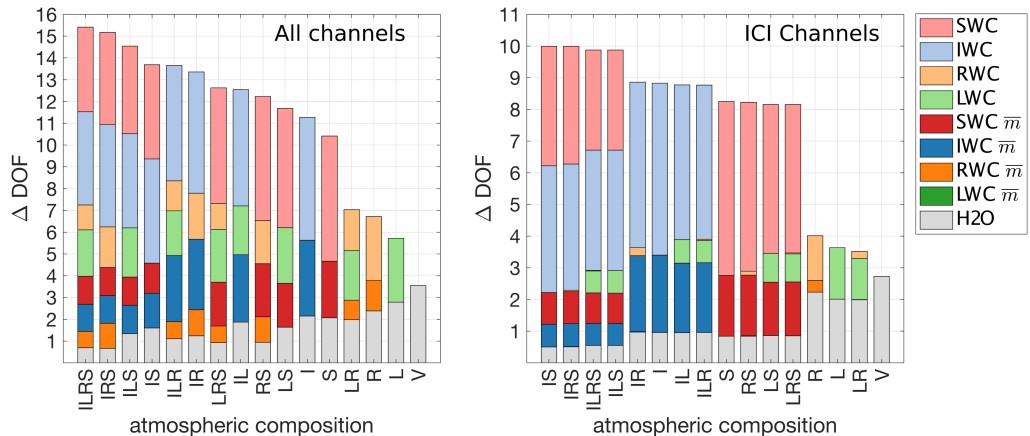

**Figure 10.** Information content $\Delta$DOF for all atmospheres, ranked after the total $\Delta$DOF. Results for the full set of channels are shown on the left, results for channels corresponding to ICI are shown on the right. Both were calculated with $\epsilon = 0.6$.



For IWC and SWC we also find some considerable information about the particle mean masses. The mean information over all cases is even higher than the mean information for H2O alone. Therefore with the chosen channels it is also possible to some extent to gain information about ice microphysics. For the liquid hydrometeors, this is hardly possible, and the mean information content for RWC is smaller than one. Especially for LWC the values are almost zero.

We now set our focus on ICI and reduce the set of channels to the 11 channels which correspond to this instrument (see Table 2). Naturally, the resulting total mean $\Delta$DOF of 7.60 is smaller than before because the number of channels is smaller, but this reduction is mostly at the cost of information about the liquid hydrometeors, not the frozen hydrometeors. For the liquid hydrometeors, on average we only get 0.94 for LWC and 0.27 for RWC. For IWC the mean information content is slightly reduced to 4.50, and for SWC to 4.25. For the particle mean masses, the one for IWC is considerably reduced to 1.51 and the

one for SWC is slightly reduced to 1.40 (Tab 4). This finding suggests that the channels below 183 GHz, which are sensitive to IWC (cp. Fig. 7), add a considerable amount to the information about the IWC mean particle mass.

In the ranking (Fig. 10) the atmospheres nicely group into four groups. The ones containing IWC and SWC build the group with highest total $\Delta$DOF, the ones containing SWC but no IWC rank second, closely followed by the ones containing IWC but no SWC. As before, the four atmospheres with the least information content are the ones not containing any frozen

hydrometeors. We also gain information about the microphysical properties of the frozen particles, although the one for IWC is considerably reduced compared to the full set of channels. The results are in agreement with previous sections, since we already found that for liquid particles the information mainly stems from the channels lower than 183 GHz and for the frozen ones from the higher channels. For the purpose of ICI it is no disadvantage to leave out the lower channels. ICI's focus is on the detection of cloud ice, and its ability to observe it on the global scale with a large spatial coverage seems to be unprecedentedly

high.

### 6.4   Realistic atmospheric profiles

So far we have only analysed one single, smooth idealised cloudy profile. To consolidate the results from the previous section, we have randomly drawn 90 more realistic cloudy profiles directly from the 10 000 ICON profiles which were used to create the mean profile (see Sect. 5.2). We calculate the information content $\Delta$DOF in the same way as before. Although an even

greater dataset would be desirable, the calculation of the Jacobians with ARTS is numerically rather expensive and we had to trade extensive statistics against computing time.

Figure 11 gives an overview of the information contents for the different hydrometeor types depending on the respective hydrometeor paths in the atmospheric column. The results from the idealised atmosphere presented above are substantiated in this statistical approach. Naturally the system tends to higher information contents for higher mass contents of the respective

hydrometeor. The values are in a similar range as we found above, except for SWC. For SWC, the $\Delta$DOF from the idealised atmosphere is amongst the outliers towards higher information contents, even though the path is well in the range of paths from the 90 atmospheres. This may be due to the fact that we took care to have all hydrometeor types in our idealised profile and that for most realistic profiles, the combination of snow and liquid clouds or rain is seldom. Thus, in general we expect to gain less information on SWC than we found for the idealised mean atmosphere.




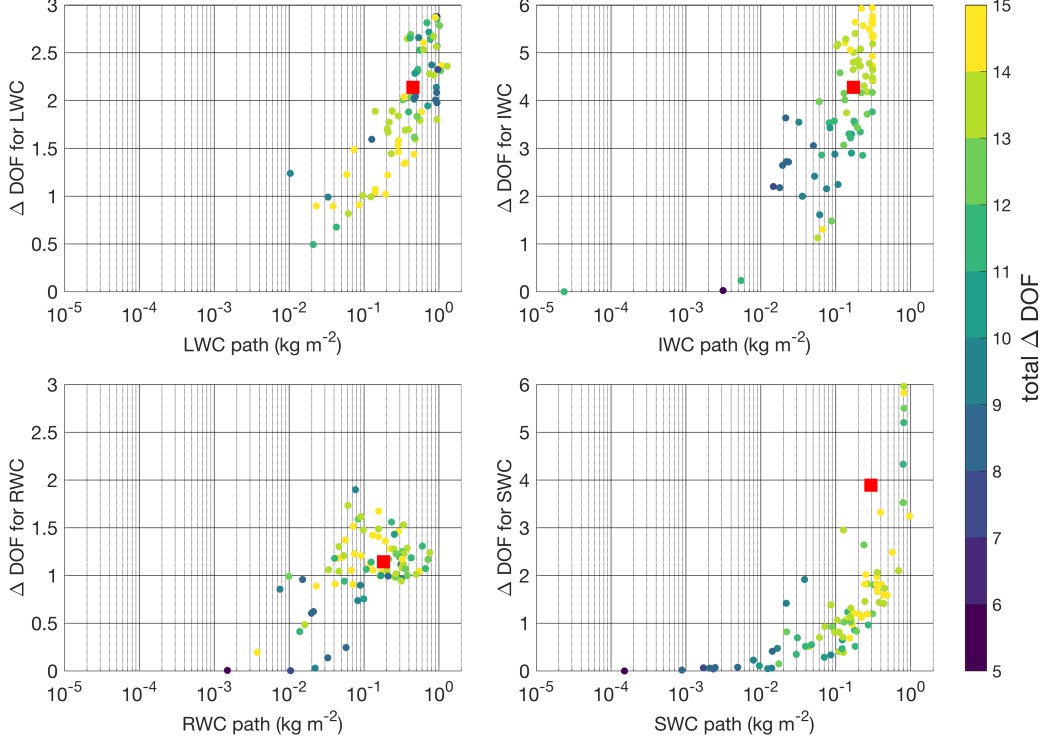

**Figure 11.** ΔDOF for the different hydrometeor mass densities over their respective column integrated path for 90 realistic atmospheres and the idealised atmosphere. The total ΔDOF is illustrated by color. The red square corresponds to the value from the idealised base profile. Note the different y axis for liquid and frozen hydrometeors.

For IWC and SWC high total ΔDOFs tend to be at high paths. For the liquid hydrometeors, the relationship between high paths and high total information content is not found. On the contrary, for LWC the low total ΔDOFs tend to be at the upper end of the LWC paths, where the cloud is mainly liquid and only little frozen water mass is present. If ice is present in the cloud, the liquid hydrometeors will be quickly consumed by the Bergeron-Findeisen process and riming, yielding lower LWC paths but higher total ΔDOFs. The most comforting result from the statistical analysis are the overall high information contents we gain for frozen hydrometeors, which again points to the ability of sensors with such high microwave channels to observe ice and snow particles in clouds on a global scale robustly regardless of the atmospheric composition.

Some caution has to be payed with regard to the physical assumptions underlying the scattering and absorption properties of ice particles. For example, Birman et al. (2017) found that changes in the size distribution and scattering properties can shift the information content from IWC to solid precipitation. Also, contrary to our study, Brath et al. (in review, 2017) did not find their retrieval being sensitive to IWC using the same channels. They base their analysis on ICON simulations with a one-moment scheme, where the size distributions for IWC and SWC are more distinct and hardly overlap, and where the IWC distribution is shifted to smaller ice particles (Sect. 3.1). Therefore the information content is distributed differently between





IWC and SWC. In nature, this arbitrary distinction between IWC and SWC does not exist and we only gain information about the whole bunch of frozen hydrometeors at once, only limited by the size and amount of the particles, and depending on their shape.

In summary, the analysis of the model atmospheres with their different compositions shows satisfactory results. Despite the strong interdependencies of the cloudy Jacobians presented in Sect. 6.2 the information content about the frozen hydrometeors proved to be high, independent of the atmospheric composition. This is especially due to the channels at high frequencies, which are only little sensitive to liquid water, and for which the Jacobians peak at different heights. Satellite missions such as ICI on MetOp SG, which employ a set of these high frequency channels therefore have a great potential to provide a robust retrieval of cloud ice and snow. For these frozen hydrometeors, contrary to the liquid ones, even an estimation of a profile may be possible, because the channels give information about different heights in the atmosphere and we get ΔDOFs of 4 to almost 5, which corresponds to 4 to 5 different heights. Also, especially for cloud ice, we consistently gain some insight into the microphysical properties, i.e., about the mean particle mass. To observe liquid hydrometeors, also lower channels such as the ones from MWI would be required. In principle we gain only little information on LWC and RWC, which is slightly dependent on the surface, which it is even decreased in the presence of cloud ice or snow. Profile retrievals for the liquid hydrometeors do not appear to be possible at all with neither setup.

## 7 Conclusions

Passive microwave instruments which employ channels higher than the well established 183.31 GHz gain more and more attention as a valuable tool to observe clouds on the global scale from space. The high frequencies especially serve to measure IWC and SWC. In fact, the upcoming MetOp-SG mission will have ICI on board, which employs channels in the range from 183.31 GHz up to 664.0 GHz.

In the past, many studies focused on the existing and well-established instruments such as AMSU-B, which measures at 89, 150 and 183.31 GHz (e.g. Hong et al., 2005, and references therein). The studies focused on the influence of the surface and of single hydrometeors on the brightness temperature spectra. Also, several studies focused on the selection of the most suitable channel sets for such instruments (Di Michele and Bauer, 2006; Jiménez et al., 2007) and on the gain of information content for hydrometeors from hyperspectral sensors (Birman et al., 2017). In this study, we performed an all-sky information content analysis for passive microwave instruments. Especially, we focused on the dependence of the information content for a certain cloud hydrometeor type on the atmospheric composition and analysed whether it is robust across the different compositions. This is worthwhile because the cloud Jacobians are highly interdependent. Also, some authors found that for example the signal of cloud ice in the 183.31 GHz channel is much weaker if the clouds precipitate compared to the signal of cloud ice from non-precipitating clouds (Greenwald and Christopher, 2002).

We chose the setups of MARSS and ISMAR, which have been flown in a recent flight campaign and complemented them by two low-frequency precipitation channels from Deimos. The resulting channels range from 23.8 GHz up to 874.4 GHz. We based our study on a high resolution simulation from the ICON model with a two-moment microphysics scheme from Seifert



and Beheng (2006). The information content was quantified by the reduction of the degrees of freedom basing on optimal estimation theory. The required Jacobians were calculated explicitly with the radiative transfer simulator ARTS (Eriksson et al., 2011; Buehler et al., 2005).

An analysis of idealised profiles containing different combinations of LWC, IWC, RWC and SWC showed that the different
Jacobians have strong interdependencies. Each component of the cloud changes the radiative background for the others, such that its presence shields or strengthens their contributions to the measured brightness temperature in the respective channel. The warming signal from H2O in the 89.0 and outermost 118.75 GHz channel is shielded by liquid hydrometeors, and the negative signal in the channels higher than 183.31 GHz turns positive in the presence of frozen hydrometeors. The signals from LWC and RWC strongly depend on the presence of other hydrometeor types and even change sign in some channels depending
on the composition of the atmosphere. The signal from frozen hydrometeors is always negative at all heights. It tends to get stronger in the presence of liquid hydrometeors, which is contrary to the findings of Greenwald and Christopher (2002) for 183.31 GHz. It slightly weakens for both if both frozen hydrometeor types, IWC and SWC, are present at the same time.

Despite these interdependencies of the Jacobians, the information content is robust with regard to the composition of the atmosphere. The information we gain about LWC is low and does not allow for profile or microphysical retrievals, which is in
accordance with findings from Crewell et al. (2009). Due to the higher channels beyond 183.31 GHz the information content on the frozen hydrometeors is high. On average ΔDOF reaches 4.99 and 4.84 for IWC and SWC. This implies a potential to retrieve profiles of the frozen hydrometeors and is due to Jacobians of the relevant channels peaking at different heights. Also, the use of these high channels enables to observe microphysical properties of IWC and SWC. Especially for IWC mean masses we find a high information content of 2.38, the one from SWC is slightly lower (1.79). However, one has to keep in
mind that the distinction between IWC and SWC in the atmospheric model is inherent in the microphysical parameterisation scheme and can not be made in reality, where the transition between the hydrometeors is continuous. Also, the model inherent microphysical size distributions influence the results. For example, the two-moment scheme used in this study tends to larger frozen hydrometeors and fewer small cloud ice particles than for example the one moment scheme from McFarquhar and Heymsfield (1997).

We have consolidated the results from the idealised profile with a set of 90 more realistic cloudy profiles from the ICON model. As expected we find a close relation between the hydrometeor path and the information we gain about that hydrometeor. The highest total information contents stem from atmospheres containing frozen hydrometeors, which is due to the fact that the scattering signal from IWC and SWC is strong, especially in the higher channels used in this study. We have to add that within these 90 profiles we find that the information content on SWC for the idealised case is amongst the outliers towards
higher information contents.

To explore the potential of ICI to observe cloud ice amount and microphysical properties on the global scale we also analysed the all-sky information content gained from that instrument. We found that the information with regard to IWC (4.50) and SWC (4.25) is only slightly lower than for the full channel set and that there is still information about the microphysical properties of the frozen particles, even though for IWC mean masses it is considerably reduced to 1.51 compared to the full channel value



of 2.38. The good performance of the ICI channel set for cloud ice and snow retrievals is very encouraging for the upcoming mission.

*Acknowledgements.*   This work was funded by the European Space Agency (ESA) under the Contract Nr. 4000113023/13/NL/MV and by the Universität Hamburg's Cluster of Excellence "Integrated Climate System Analysis and Prediction" (CliSAP, funded by DFG). The authors
5   thank the Max-Planck Institute for Meteorology, Hamburg, and the project HD(CP)$^2$ for providing the atmospheric model data for this study and the ARTS community for providing and developing the radiative transfer model ARTS. We thank Axel Seifert for his support with regard to the two-moment microphysics scheme, Rémy Roca and Jean-Franqis Mahfouf for the very valuable scientific discussions and support, and Oliver Lemke for technical support of the study.



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
