# Peer review of "All-sky Information Content Analysis for Novel Passive Microwave Instruments in the Range from 23.8 GHz up to 874.4 GHz"

_Atmospheric Measurement Techniques, 2017_

## Referee Comment (RC1) · Anonymous Referee #2 · 5 Jan 2018

This is a comprehensive study on the idealized information content from microwave/sub-millimeter microwave channels that are relevant to the current instruments deployed in field and space missions. I have three major comments: 1. I agree with the author that it is highly important to have consistent micro-physical parameterisations in the RTM and atmospheric model and appreciate the careful discussions on the comparisons between Seifer and Beheng 2006 and McFarquhar and Heymsfield (1997) schemes. However, the McFarquhar and Heymsfield (1997) parameterization is developed for tropical cirrus cloud using field campaign data collected during CEPEX, which may not be proper to apply to a midlatitude frontal cloud system. Besides, I don't see why it is necessary to have such long discussions in this article if two-moment

scheme is used in both ICON and ARTS. 2. In the calculation of Jacobians, the channel response function is not used and instead monochromatic radiative transfer simulations for the center frequencies of the side bands are carried out. For channels in the window region and sounding channels far from the absorption line ceter, the sensitivities or information content are sensitive to the width of the channel. And these channels are used to retrieve the hydrometeors. 3. P10. Line 8: please explain in more detail: "the scattering solver for the perturbations gets the reference result as a first guess". Scattering is important since the focus of this study is to understand the information content in these channels to the different combination and types of hydrometers.

Minor comments: 1. In the abstract, Line 14: "however the information content is robust", this is right after the discussion on the little information on the profiles and microphysics. "robust" with respect to what? 2. P2, Line 34: suggests to change to "low level clouds have only little effect on the . . ." 3. P3. Line 25: remove comma in 183GHz. 4. P3. Line 34: add "in" before "Sect.2". 5. P4. Line 24: Suggest to remove the first sentence in this paragraph, and state what kind of assumptions are made for surface emissivity and surface type. 6. P7, Line 4: "smaller smaller" 7. P12, Line 19: "to choose them". Also, should it be "for each hydrometeor type"? Line 18: "amongst the extremes": does this mean extreme profiles are selected? If so, it is contradict with following statement that outliers are excluded. Please clarify. 8. P24, Line 8: "has to be paid"

---

## Referee Comment (RC2) · Anonymous Referee #1 · 15 Jan 2018

**Review of manuscript: 'All-sky Information Content Analysis for Novel Passive Microwave Instruments in the Range from 23.8 GHz up to 874.4 GHz' by V. Grützun et al.**

**General**

This paper attempts to address issues related to the information content of space-borne microwave and sub-millimeter observations of clouds and precipitation. In particular it uses linear optimal estimation theory to quantify the information content of novel space-borne sensors that currently are being developed (ICI) and of existing airborne sensors (ISMAR and MARSS).

This is an interesting, timely, and relevant topic. However, unfortunately, this paper has a couple of fundamental flaws and shortcomings and I cannot recommend publication at this point. Beyond this, I am also somewhat skeptical of the relevance and global applicability of the results presented, even if the flaws were corrected. My major criticisms are as follows.

**1. Major comments**

**1.1. The calculation of the a-priori covariance matrix $S_a$ is flawed**

Calculating Sa in log-space for positive semi-definite quantities such as SWC poses the challenge of what to do with the zeros, as correctly stated by the authors. However, their approach of setting zeros to 2.22E-16 to numerically avoid this issue will have a significant impact on Sa and DOF.

A simple example: Assume a quantity (e.g. SWC) for which we have two-hundred values (e.g. SWC at 500 hPa from 200 profiles).  Say, one-hundred of these values are zeros ('cloud-free') and one-hundred are  0.1 (in some appropriate units). What is the variance of log(SWC) now? If I set the zeros to 2.22E-16, take the logarithm of this two-hundred element vector, and calculate its variance, the variance comes out to be  ~286. But why chose 2.22E-16, would not a good approximation for zero be 1E-10? If I do that, the variance is reduced to ~107. If I assume 1E-8 is zero, then the variance becomes ~65. Or, maybe I should assume double-precision? In that case I can set the zero SWC values to, say, 1D-50 and the variance becomes ~3200. Whichever number it is, this number will populate the diagonal elements of Sa.

So, ultimately Sa becomes arbitrary.

This issue is only slightly ameliorated if there are fewer 'cloud-free' observations that have to be replaced because whichever small number I chose to fill in the zeros, they will constitute outliers and dominate the variance and thereby Sa. This also explains the authors statement that '*The covariance [...] goes up to 670 in units of the natural logarithm on the diagonal of the SWC–SWC block matrix*'.

This will have tremendous effects on the value of DOF. The larger Sa becomes, the larger DOF will be. That is, the less the a-priori is constrained, the more influence the

measurements will have. Below is an example of this using a very simple fictional observing system with two observations and two elements of the state space (and identity Jacobians, which have no impact on the principal point made here.)

Again, the point is: The magnitude of Sa relative to Sy will have a very strong impact on DOF. The larger the values of Sa, the higher DOF will be. If the choice of Sa is arbitrary, the resulting DOF will be arbitrary. This issue in itself invalidates the paper results.

[Figure]

Plot shows the strong dependence of DOF on the magnitude of Sa relative to Sy.

```
; IDL Code to create above plot…

; create some values between 0.01 and 1000. for Sa
sa_mag = 10.0^(FINDGEN(51)/10.-2)

; will hold results
dof    = sa_mag* 0.0

; obs error covariance and inverse (identity matrix) .. does not change
sy     = DIAG([1D,1D])
sym1   = INVERT(sy)

; go through DOF math for all values of Sa...
FOR i=0,N_ELEMENTS(sa_mag)-1 DO BEGIN
  sa     = DIAG([sa_mag[i],sa_mag[i]]) ; create 2x2 diagonal matrix Sa
  sam1   = INVERT(sa)
  sr     = INVERT(sam1+sym1)
  dof[i] = TRACE(diag([1D,1D])-sr ## sam1)
ENDFOR

; plot results
p = PLOT(sa_mag,dof,XTITLE='Variance (diag. elements of Sa)',YTITLE='DOF',/XLOG,THICK=2,FONT_SIZE=14)
p.save,'test.png'

END
```

**1.2. Too small variability in underlying dataset**

This study is based on a very limited set of model data (one mid-latitude frontal event). While the sensitivity study regarding the one average profile and the 90 selected profiles does show some variability, tropical, mid-latitude wintertime and other situations are simply not captured by this study. These situations will not only likely have dramatically different a-priori covariances, but also dramatically different Jacobians. For example in a very dry atmosphere, the sounding capabilities at higher frequencies will be reduced as more and more channels might see further down in the atmosphere. This will decrease DOF. Similarly, in a very intense tropical deep convective area, nearly all weighting functions that peak in the mid- and lower troposphere will move up because the atmosphere becomes optically very thick. This, again, will increase redundancy and reduce DOF. While the authors acknowledge the shortcomings of the limited dataset, none of these effects is quantified or even discussed.

So, even if the methodology was right and Sa was calculated correctly, the results will be of very limited use in characterizing the instruments.

**1.3. Key concept for lower frequencies missing**

A key novel concept of the Metop-SG constellation is the combination of the 118 GHz and 50-60 GHz oxygen sounding channels for precipitation retrievals as outlined for example in Bauer and Mugnai (2003)[1]. This aspect is completely ignored in the current study and only a reduced set of three channels below 118GHz is even considered, none of which are sounding channels. Therefore, the authors conclusion that '*The information about the liquid hydrometeors comes from the lower channels and is comparably low (2.36 for liquid cloud water and 1.81 for rain).*' appears to not be justified.

A fair assessment of this statement with regard to Metop-SG would have to include the full set of MWI channels sounding channels. For the case of the airborne MARSS system, the finding is probably correct, but given there are only three low-frequency channels it is no surprise the information content comes out to be somewhere between one and three. This is also consistent with the existing large body of literature on low-frequency precipitation and cloud liquid water retrievals.

I suggest either this is addressed in full (including the 50-60 sounding channels), or at the very least much more emphasis is put on this aspect and/or the very limited nature of this particular finding be highlighted.

**2. Other comments**

**Page 5, Line 19/20**: "*It is crucial to match the microphysical parameterisations of the radiative transfer model with those of the atmospheric model.*" I do not agree
* * *
[1] Bauer and Mugnai: JGR, VOL. 108, NO. D23, 4730, doi:10.1029/2003JD003572, 2003

with this. It would be perfectly fine to use for example different habits that are not consistent with the assumptions made in the ICON microphysics parameterizations, e.g. in the m-D relationship. The variability imposed by ice habits on the simulations (and thereby also on Sy) is not discussed. Forward model errors are not accounted for in general in Sy, which only seems to account for reasonable estimates for instrument noise (1 K).

**Page 10, line 8-9**: "*Instead, the scattering solver for the perturbations gets the reference result as a first guess, which saves most of the iterations that would otherwise be needed.*" Why first guess? I do not understand. Needs more explanation.

**E. g. Figure 11**: Use of term "LWC Path" etc is confusing… Should be LWP ('Liquid Water Path'). In general the distinction between 'content' and 'path' is somewhat blurry in the paper. The authors jump between the two but consistently use e.g. LWC. The impact of what the authors call 'shielding' is much better understood in terms of path integrated properties. For example, for 'shielding' it matters how much ice in total (in kg/m$^2$) is above the liquid, whereas IWC (in kg/m$^3$) is only of secondary importance. This should be made clearer and the discussion should be expanded.

**Page 14, near Figure 3 or Table 3**: Please provide column integrated values of LWP, IWP, SWP, and H2O and RWP….. This would be very helpful in getting a feeling for the atmosphere.

**(Page 25, Lines 17)** to **(Page 26, line 3)** are largely just a repetition of the introduction and other parts of Section 2. Should be removed.

**Page 26, Line 6:** '…*its presence shields or strengthens*….' Instead of 'shields vs strengthens maybe use increases/decreases or weakens/strengthens (something 'shielding' something else could, I presume, by used as the explanation for why a weakening occurs in this context.

**3. Minor comments**

Page 3, Line 34: **in** Sec. 2..

Page5, line 4  I suggest 'are *somewhat* smaller…'

Page 25, Line rephrase 'whole bunch' with 'sum of the two' or something similar.

---

## Author Comment (AC1) · 27 Apr 2018

First of all. A heartfelt thank you for the thorough review of the paper. We very much appreciate the time and effort reviewer 1 put into the review, and we believe that he or she enabled a considerable improvement of the article, especially with regard to the choice of our a priori error covariance. In the following we address the reviewer's comments in detail. All changes are highlighted in a track changes version of the manuscript.

**Initial Review**

1. there are many grammar mistakes. These should be corrected before accepting the manuscript.

We have given the manuscript to a native English speaker to correct the mistakes and included his corrections.

2. Some conclusions reported in the abstract are not supported by the results. Such as for frozen hydrometers "Profile retrievals may be possible for the mass densities and some information about the microphysical properties, especially for cloud ice, can be gained." and for liquid hydrometer "There is little information about the profile or the microphysical properties".

We have thoroughly revised the definition of the a priori error covariance, which changed the results (see below). We adjusted the abstract accordingly.

Review of manuscript: 'All-sky Information Content Analysis for Novel Passive Microwave Instruments in the Range from 23.8 GHz up to 874.4 GHz' by V. Grützun et al.

**General**

This paper attempts to address issues related to the information content of space-borne microwave and sub-millimeter observations of clouds and precipitation. In particular it uses linear optimal estimation theory to quantify the information content of novel spaceborne sensors that currently are being developed (ICI) and of existing airborne sensors (ISMAR and MARSS). This is an interesting, timely, and relevant topic. However, unfortunately, this paper has a couple of fundamental flaws and shortcomings and I cannot recommend publication at this point. Beyond this, I am also somewhat skeptical of the relevance and global applicability of the results presented, even if the flaws were corrected. My major criticisms are as follows.

**1. Major comments**

**1.1. The calculation of the a-priori covariance matrix Sa is flawed**

Calculating Sa in log-space for positive semi-definite quantities such as SWC poses the challenge of what to do with the zeros, as correctly stated by the authors. However, their approach of setting zeros to 2.22E-16 to numerically avoid this issue will have a significant impact on Sa and DOF.

A simple example: Assume a quantity (e.g. SWC) for which we have two-hundred values (e.g. SWC at 500 hPa from 200 profiles). Say, one-hundred of these values are zeros ('cloud-free') and one-hundred are 0.1 (in some appropriate units). What is the variance of log(SWC) now? If I set the zeros to 2.22E-16, take the logarithm of this two-hundred element vector, and calculate its variance, the variance comes out to be ~286. But why chose 2.22E-16, would not a good approximation for zero be 1E-10? If I do that, the variance is reduced to ~107. If I assume 1E-8 is zero, then the variance becomes ~65. Or, maybe I should assume double-precision? In that case I can set the zero SWC values to, say, 1D-50 and the variance becomes ~3200. Whichever number it is, this number will populate the diagonal elements of Sa. So, ultimately Sa becomes arbitrary.

This issue is only slightly ameliorated if there are fewer 'cloud-free' observations that have to be replaced because whichever small number I chose to fill in the zeros, they will constitute outliers and dominate the variance and thereby Sa. This also explains the authors statement that 'The covariance [...] goes up to 670 in units of the natural logarithm on the diagonal of the SWC-SWC block matrix'.

This will have tremendous effects on the value of DOF. The larger Sa becomes, the larger DOF will be. That is, the less the a-priori is constrained, the more influence the measurements will have. Below is an example of this using a very simple fictional observing system with two observations and two elements of the state space (and identity Jacobians, which have no impact on the principal point made here.)

Again, the point is: The magnitude of Sa relative to Sy will have a very strong impact on DOF. The larger the values of Sa, the higher DOF will be. If the choice of Sa is arbitrary, the resulting DOF will be arbitrary. This issue in itself invalidates the paper results.

 $\textit{Figure 1 Plot shows the strong dependence of DOF on the magnitude of Sa\ relative\ to\ Sy.}$

We are very grateful for this comment, and for the effort you have put into this point! We agree with your arguments and have re-evaluated the definition of our a priori covariance (see entire Section 4.2 "A priori covariance"). In fact, the new definition gives far more physical results for the information content. Especially, we now get some ability to detect liquid clouds and rain better (Fig. 10 and 12, Tab. 4), which, on a second thought, makes much

more sense, because our set of channels includes five channels in the 118 GHz region, and one at 89, 50, and 23 GHz each. These channels are indeed sensitive to liquid hydrometeors. Also, as you pointed out, Bauer and Mugnai showed that an extended (compared to the instruments we use) set of channels around the 118 GHz oxygen line and within the oxygen absorption complex region between 50 and 57 GHz gives good results for profile retrievals of precipitation. We do not expect equally good results in our case, because we would need more lines within the absorption complex around 50 GHz, but we indeed should have paid more attention to the fact that the information content for liquid cloud and rain was so low.

We are aware that the choices we make for our a priori matrix will have an influence on the variance and therefore on the information content. This is the case in each optimal estimation retrieval, and finding a suitable a priori covariance is one of the challenges in doing such retrievals. For the revised version of the article, we have employed more reasonable and justifiable thresholds as follows, and we also discuss the influence of the choices in more detail in the result section (see below).

- 1. We assume that we have a working algorithm to detect clouds in the first place and only include cloudy columns. The criterion for "cloudy" for the profiles of the ICON simulation is that the total condensed water path exceeds 1e-4 kg m-2. This threshold, however, does not affect the results much. Furthermore, we include all available time steps of the simulation, such that the number of profiles for the calculation is increased (we chose the cloudiest time step in the initial article draft).
- 2. We now clip the values instead of only setting the zeros to our thresholds. Clipping in our sense means that any value, which is smaller than the threshold value is set to the respective threshold. With the very small numerical threshold, this did not make much of an effect, but with the larger thresholds (see below), this has a great influence on the variance of the system.
- 3. For the mass densities, we employed a threshold of 1e-7 kg m-3. Assuming a detection limit for, e.g., cloud ice water path of 1 g m-2 (see e.g. Brath et al., 2018), and a cloud thickness of about 2.5 km, we end up with a cloud ice mass density threshold in the order of 1e-7 kg m-3. This also approximately corresponds to a numerical threshold within the two-moment scheme (approximately, because the scheme employs mass mixing ratios instead of mass densities). Beyond this value, for example collisional processes can take place.
- 4. For the mean particle masses, we employed the intrinsic lower thresholds of the two-moment scheme. We have experimented with setting the threshold in dependence of the threshold for the mass densities, but this introduced artificial correlations between the two quantities which were not present in the first place.

With the chosen thresholds, the main peaks of the distribution of the hydrometeor mass densities (and by definition also of the mean masses) are considered for the calculation of the a priori covariance. These main peaks are the physically meaningful values within the simulation and therefore should be, and with the chosen thresholds are, included in the calculation.

The threshold for the mass densities has the biggest influence. We have explored the dependency of the mean information content on this particular threshold (see Figure 2, which is also included in the revised manuscript as Fig. 11). The principle relations between the information content for the mass densities stay the same, but naturally the overall information content decreases for an increasing threshold. The information content for the mean masses on the other hand somewhat increases. This is based on the combined retrieval and influenced by the cross correlations between the mass densities and the mean masses, since the threshold for the mean masses stays constant over the whole plot. The chosen threshold of 1e-7 for the mass density gives reasonable results for the information content.

Figure 2 Information content, dependence on chosen threshold. Left:  $\Delta DOF$  for mass densities. Right:  $\Delta DOF$  for mean masses. The threshold for the mass densities is altered, the one for the mean masses stays constant. The line shows the mean information content, the shaded area lies between the minimum and maximum information content. The dependency of the information content of the mean masses on the threshold for the mass densities is due to the covariance or the respective cross correlations of mass densities and mean masses.

The new a priori covariance matrix first is more consistent with the microphysical scheme on the one hand, second has more physical based thresholds to clip small values from the profiles, and third also gives more reasonable results for the information content. We have included some discussion about the dependencies in the manuscript (mainly p24, IS - p25, IS

**1.2. Too small variability in underlying dataset**

This study is based on a very limited set of model data (one mid-latitude frontal event). While the sensitivity study regarding the one average profile and the 90 selected profiles does show some variability, tropical, mid-latitude wintertime and other situations are simply not captured by this study. These situations will not only likely have dramatically different a-priori covariance, but also dramatically different Jacobians. For example, in a very dry atmosphere, the sounding capabilities at higher frequencies will be reduced as more and more channels might see further down in the atmosphere. This will decrease DOF. Similarly, in a very intense

tropical deep convective area, nearly all weighting functions that peak in the mid- and lower troposphere will move up because the atmosphere becomes optically very thick. This, again, will increase redundancy and reduce DOF. While the authors acknowledge the shortcomings of the limited dataset, none of these effects is quantified or even discussed.

So, even if the methodology was right and Sa was calculated correctly, the results will be of very limited use in characterizing the instruments.

Yes, the use of one case limits the study to some extent. We believe that the basic principles can be made clear on the basis of this case, especially the interdependencies of the Jacobians. We acknowledge that the Jacobians will look different in different regimes and have added some phrases regarding this topic to the discussion (p4, l10-14).

**1.3. Key concept for lower frequencies missing**

A key novel concept of the Metop-SG constellation is the combination of the 118 GHz and 50-60 GHz oxygen sounding channels for precipitation retrievals as outlined for example in Bauer and Mugnai (2003)1. This aspect is completely ignored in the current study and only a reduced set of three channels below 118GHz is even considered, none of which are sounding channels. Therefore, the authors conclusion that 'The information about the liquid hydrometeors comes from the lower channels and is comparably low (2.36 for liquid cloud water and 1.81 for rain).' appears to not be justified. A fair assessment of this statement with regard to Metop-SG would have to include the full set of MWI channels sounding channels. For the case of the airborne MARSS system, the finding is probably correct, but given there are only three low-frequency channels it is no surprise the information content comes out to be somewhere between one and three. This is also consistent with the existing large body of literature on lowfrequency precipitation and cloud liquid water retrievals.

I suggest either this is addressed in full (including the 50-60 sounding channels), or at the very least much more emphasis is put on this aspect and/or the very limited nature of this particular finding be highlighted.

The focus of our study was more on the instruments ICI and ISMAR, complemented by a few lower frequency channels from the instrumentation which has been flown on the FAAM aircraft. We did not intend to put our focus on the full microwave suite of Metop-SG. We should have stated that clearer in the initial manuscript, and apologize for the oversight! We now have put more emphasis on the fact that we do not expect to gain a large amount of information about liquid hydrometeors with the channels, which we employ (e.g. p22, I3 – 9, again stressed in conclusions p28, I15-17). We have emphasized that our focus is more on the detection of frozen hydrometeors with the instruments ICI and ISMAR. However, with the newly defined a priori error covariance, liquid water gains a greater proportion of the total information content, which stems from the 23, 89, 50 and the outer 118 GHz lines (Fig. 10, Tab. 4). It seems that the initially defined a priori error favoured frozen hydrometeors more than liquid ones. We changed our discussion accordingly and more thoroughly included the fact that more channels in the lower frequency regions are needed to retrieve liquid hydrometeor retrievals and we stressed that precipitation retrievals with these low channels are established techniques (as above, p22, I3 - 9). We have included the information that such

low frequency channels will be available on the Metop-SG satellite (p27, l14 - 17). We also have included the reference Bauer and Mugani, 2003.

**2. Other comments**

Page 5, Line 19/20: "It is crucial to match the microphysical parameterisations of the radiative transfer model with those of the atmospheric model." I do not agree with this. It would be perfectly fine to use for example different habits that are not consistent with the assumptions made in the ICON microphysics parameterizations, e.g. in the m-D relationship. The variability imposed by ice habits on the simulations (and thereby also on Sy) is not discussed.

Thank you for pointing this out. You are right, the m-D relationship in the microphysical model is only implicitly used, e.g. for the calculation of fall speeds. Therefore, an inconsistency with regard to this parameter would probably be less crucial than an inconsistency in the employed size distributions. We still would like to use as much information from the two-moment scheme as possible to perform our analysis. We have included some discussion about this in lines (p6, 122 - p.7, 13).

Forward model errors are not accounted for in general in Sy, which only seems to account for reasonable estimates for instrument noise (1 K).

This is true. Within the scope of this study, we assume we have a perfect forward operator. We have added the information in line (p8, 129 - p9, 12).

Page 10, line 8-9: "Instead, the scattering solver for the perturbations gets the reference result as a first guess, which saves most of the iterations that would otherwise be needed." Why first guess? I do not understand. Needs more explanation.

For clarity, we have rephrased this explanation as follows:

"In practice, we do not make a fully independent  $T_B$  calculation for each perturbation, since this is computationally very inefficient for the iterative scattering solver used (Emde et al., 2004). Instead, the scattering solver uses the result from the unperturbed scheme as a starting point. That result should be close to the result from the perturbed case already, because our profile perturbations are small. From that starting point, the perturbed Jacobians are calculated with far fewer iterations compared to a completely uneducated starting point, which makes the scheme far more computationally efficient." (p10, l11-16)

E. g. Figure 11: Use of term "LWC Path" etc is confusing... Should be LWP ('Liquid Water Path'). In general the distinction between 'content' and 'path' is somewhat blurry in the paper. The authors jump between the two but consistently use e.g. LWC.

We have included the term "LWP", "IWP", ... for the liquid water path, ice water path, ... (p5, l19-20) and use it more consistently throughout the article, including a corrected Figure 11.

<sup>1 Bauer and Mugnai: JGR, VOL. 108, NO. D23, 4730, doi:10.1029/2003JD003572, 2003

The impact of what the authors call 'shielding' is much better understood in terms of path integrated properties. For example, for 'shielding' it matters how much ice in total (in kg/m2) is above the liquid, whereas IWC (in kg/m3) is only of secondary importance. This should be made clearer and the discussion should be expanded.

We have expanded the discussion (line mainly p16, l8-15). We acknowledge the fact that the paths in combination with the sensitivity of the respective channels in the regions where hydrometeors reside are the main contributors to the signal at the top of the atmosphere. Also, we agree that the term "shielding" implies that a hydrometeor below a region with high H2O or a large path of another hydrometeor such as cloud ice, is hidden. We introduced the term to imply that H2O or another hydrometeor between the one to be detected and the sensor is not seen because the sensor can't penetrate the atmosphere down to the hydrometeor to be detected. We now use the term "shielding" more sparsely and swapped it for "weakening" where applicable.

Part of the weakening of the signal is also due to the specific radiative background, which the other hydrometeors and H2O create. Depending on the radiative background, the signal also might strengthen. For example, in Figure 7 it is evident that the Jacobians for H2O change much in exactly those regions where the cloud ice is located and the cloud ice Jacobians peak. This implies that it is not only the atmospheric part above the regarded component, which alters the signal from that component.

Page 14, near Figure 3 or Table 3: Please provide column integrated values of LWP, IWP, SWP, and H2O and RWP..... This would be very helpful in getting a feeling for the atmosphere.

We have provided the values in the figure caption of Figure 3.

(Page 25, Lines 17) to (Page 26, line 3) are largely just a repetition of the introduction and other parts of Section 2. Should be removed.

We have considerably shortened the paragraph. We would like to keep at least a small introduction in the conclusions and hope that the shortened version is acceptable.

Page 26, Line 6: '...its presence shields or strengthens....' Instead of 'shields vs strengthens maybe use increases/decreases or weakens/strengthens (something 'shielding' something else could, I presume, by used as the explanation for why a weakening occurs in this context.

We have changed to "weakens/strengthens".

**3. Minor comments**

Page 3, Line 34: in Sec. 2..

We have corrected this.

**Page5, line 4 I suggest 'are somewhat smaller...'**

We didn't find a phrase matching this comment in that line on p. 5. Did you mean p. 7 l. 4 "the largest snow hydrometeors are little bit smaller smaller than in the two-moment scheme"? We have completely altered that section and the respective phrase doesn't exist anymore.

**Page 25, Line rephrase 'whole bunch' with 'sum of the two' or something similar.**

We have rephrased to: "we gain information about the whole set of frozen hydrometeors".

---

## Author Comment (AC2) · 27 Apr 2018

We thank Referee #2 for his or her time and effort, and for the very valuable comments. We have clarified and corrected the manuscript accordingly (the individual comments are addressed below). Note that due to some major comments of Referee #1 there were major changes in the manuscript. Especially, Referee #1 critisised the choices we made for the calculation of the a priori covariance error from ICON. We therefore recalculated it with more consistent and physically based assumptions. This changes the results for the information content. The general conclusions remain valid, but the liquid phase gains a greater share of the overall information content.

This is a comprehensive study on the idealized information content from microwave/sub-millimeter microwave channels that are relevant to the current instruments deployed in field and space missions.

**I have three major comments:**

1. I agree with the author that it is highly important to have consistent micro-physical parameterisations in the RTM and atmospheric model and appreciate the careful discussions on the comparisons between Seifer and Beheng 2006 and McFarquhar and Heymsfield (1997) schemes. However, the McFarquhar and Heymsfield (1997) parameterization is developed for tropical cirrus cloud using field campaign data collected during CEPEX, which may not be proper to apply to a midlatitude frontal cloud system. Besides, I don't see why it is necessary to have such long discussions in this article if two-moment scheme is used in both ICON and ARTS.

Thank you for this very valuable suggestion. We have reorganized the discussion of the microphysics. We have much more focused on the two-moment physics and removed the major part of the discussion of the McFarquhar and Heymsfield scheme, including the respective figure. We left a paragraph in, which points to the difference between the two schemes, but we added the information that the one-moment scheme is developed for the tropics (Sec. 3.1, esp. p6, l15-21).

2. In the calculation of Jacobians, the channel response function is not used and instead monochromatic radiative transfer simulations for the center frequencies of the side bands are carried out. For channels in the window region and sounding channels far from the absorption line center, the sensitivities or information content are sensitive to the width of the channel. And these channels are used to retrieve the hydrometeors.

We have added the following to the discussion:

"We do not use an explicit sensor response function but perform monochromatic radiative transfer simulations for the center frequencies of the side bands in each channel and use the mean of the two brightness temperatures. For clear sky, highly resolved (in terms of frequencies) tests showed that the error compared to using one monochromatic frequency per side band is less than 1 K (Brath et al., 2018). Because the scattering properties of the hydrometeors change only marginally within the band width, a further increase of this uncertainty in the cloudy case is unlikely."

Brath, M., Fox, S., Eriksson, P., Harlow, R. C., Burgdorf, M., and Buehler, S. A.: Retrieval of an IceWater Path over the Ocean from ISMAR and MARSS millimeter/submillimeter brightness temperatures, Atmos. Meas. Tech. Discuss., doi:10.5194/amt-2017-167, 2018.

3. P10. Line 8: please explain in more detail: "the scattering solver for the perturbations gets the reference result as a first guess". Scattering is important since the focus of this study is to understand the information content in these channels to the different combination and types of hydrometers.

For clarity, we have rephrased this explanation as follows:
"In practice, we do not make a fully independent $T_B$ calculation for each perturbation, since this is computationally very inefficient for the iterative scattering solver used (Emde et al., 2004). Instead, the scattering solver uses the result from the unperturbed scheme as a starting point. That result should be close to the result from the perturbed case already, because our profile perturbations are small. From that starting point, the perturbed Jacobians are calculated with far fewer iterations compared to a completely uneducated starting point, which makes the scheme far more computationally efficient." (p10, l11-16)

**Minor comments:**

1. In the abstract, Line 14: "however the information content is robust", this is right after the discussion on the little information on the profiles and microphysics. "robust" with respect to what?
We have rephrased to: "…the information content is surprisingly robust across different atmospheric compositions."

2. P2, Line 34: suggests to change to "low level clouds have only little effect on the "
We have rephrased to: "Low level clouds have only a marginal effect on…"

3. P3. Line 25: remove comma in 183GHz.
We have corrected this.

4. P3. Line 34: add "in" before "Sect.2".
We have corrected this and rephrased slightly.

5. P4. Line 24: Suggest to remove the first sentence in this paragraph, and state what kind of assumptions are made for surface emissivity and surface type.
We have rephrased the paragraph to:

"The radiative transfer simulations were performed with two different surface emissivities $\varepsilon$. In the first set of simulations, $\varepsilon$ is equal to 0.6, which corresponds to an ocean surface. In the second set of simulations, $\varepsilon$ is equal to 0.9, which corresponds to a land surface. Further, specular reflection is assumed. One should keep in mind though, that in reality $\varepsilon$ depends strongly on the specific surface and to a smaller extent on the channel. However, the results differ only little for the different emissivities. Therefore, we use the simplified assumption of a constant emissivity for all channels, and the main part of the results shown in this article will be for the emissivity of the ocean, i.e., $\varepsilon$=0.6." (p5, l3-8)

6. P7, Line 4: "smaller smaller"
The respective section has changed much, and this phrase does not exist anymore.

7. P12, Line 19: "to choose them". Also, should it be "for each hydrometeor type"?
We have corrected both and slightly rephrased (now p13, l18-19).

Line 18: "amongst the extremes": does this mean extreme profiles are selected? If so, it is contradict with following statement that outliers are excluded. Please clarify.
We have rephrased to:
"To exclude unphysical outliers, which may be produced by the model, we disregard the profiles with a path larger than the 99$^{th}$ percentile." (p13, l19-20) Occasionally unphysical values may appear due to numerical issues. In order to only include valid profiles, we exclude the upper percentage of the extreme profiles.

8. P24, Line 8: "has to be paid"
We have corrected this (now p26, l11).